# Gamete expression of TALE class HD genes activates the diploid sporophyte program in *Marchantia polymorpha*

**Tom Dierschke[1,2], Eduardo Flores-Sandoval[1], Madlen I Rast-Somssich[1], Felix Althoff[2], Sabine Zachgo[2], John L Bowman[1]***

[1]School of Biological Sciences, Monash University, Melbourne, Australia; [2]Botany Department, University of Osnabrück, Osnabrück, Germany

**Abstract** Eukaryotic life cycles alternate between haploid and diploid phases and in phylogenetically diverse unicellular eukaryotes, expression of paralogous homeodomain genes in gametes primes the haploid-to-diploid transition. In the unicellular chlorophyte alga *Chlamydomonas*, KNOX and BELL TALE-homeodomain genes mediate this transition. We demonstrate that in the liverwort *Marchantia polymorpha,* paternal (sperm) expression of three of five phylogenetically diverse BELL genes, Mp*BELL234*, and maternal (egg) expression of both Mp*KNOX1* and Mp*BELL34* mediate the haploid-to-diploid transition. Loss-of-function alleles of Mp*KNOX1* result in zygotic arrest, whereas a loss of either maternal or paternal Mp*BELL234* results in variable zygotic and early embryonic arrest. Expression of Mp*KNOX1* and Mp*BELL34* during diploid sporophyte development is consistent with a later role for these genes in patterning the sporophyte. These results indicate that the ancestral mechanism to activate diploid gene expression was retained in early diverging land plants and subsequently co-opted during evolution of the diploid sporophyte body.

*For correspondence:
John.Bowman@monash.edu

**Competing interest:** The authors declare that no competing interests exist.

## Introduction

Life cycles of eukaryotes alternate between haploid and diploid phases, initiated by meiosis and gamete fusion, respectively. Expression of paralogous homeodomain (HD) genes in the two gametes and the subsequent heterodimerization of the respective proteins in the zygote direct the haploid-to-diploid transition in gene expression in phylogenetically diverse eukaryotes, including the ascomycete fungus *Saccharomyces cerevisiae* (**Goutte and Johnson, 1988**; **Herskowitz, 1989**), the basidiomycete fungi *Coprinopsis cinerea* and *Ustilago maydis* (**Gillissen et al., 1992**; **Hull et al., 2005**; **Kues et al., 1992**; **Spit et al., 1998**, **Urban, 1996**), the Amoebozoa *Dictyostelium discoideum* (**Hedgethorne et al., 2017**), the brown alga *Ectocarpus* (**Arun et al., 2019**), the red alga *Pyropia yezoensis* (**Mikami et al., 2019**), and the unicellular chlorophyte alga *Chlamydomonas reinhardtii* (**Ferris and Goodenough, 1987**; **Lee et al., 2008**; **Nishimura et al., 2012**; **Zhao et al., 2001**). This broad phylogenetic distribution suggests this was an ancestral function of HD genes (reviewed by **Bowman et al., 2016b**). In Viridiplantae, the paralogs are two subclasses, KNOX and BELL, of TALE class HD genes. In *Chlamydomonas*, the minus (-) gamete expresses a KNOX protein (GSM1) and the plus (+) gamete expresses a BELL protein (GSP1), and upon gamete fusion, the two proteins heterodimerize and translocate to the nucleus, activating zygotic gene expression (**Lee et al., 2008**). GSM1 and GSP1 are necessary for diploid gene expression and, when ectopically expressed together in vegetative haploid cells, are sufficient to induce the diploid genetic program (**Ferris and Goodenough, 1987**; **Lee et al., 2008**; **Nishimura et al., 2012**; **Zhao et al., 2001**). Biologically, the expression of a unique paralog in each type of gamete, coupled with the requirement for heterodimerization for functionality, is a

mechanism to preload gametes such that immediately following gamete fusion/fertilization a distinct diploid genetic program is initiated.

Heterodimerization of TALE-HD paralogs is mediated by subclass-specific protein domains N-terminal of the homeodomain conserved in phylogenetically diverse eukaryotes (*Bellaoui et al., 2001*; *Bürglin, 1997*). For example, Viridiplantae KNOX proteins and metazoan MEIS proteins share a homologous heterodimerization domain, thus defining a TALE-HD subclass, named MEINOX, present in the ancestral eukaryote (*Bürglin, 1997*). In contrast, heterodimerization domains of MEINOX partners are not as well conserved (*Joo et al., 2018*). Some MEINOX partners in animals have a characteristic PBC-C heterodimerization domain (*Bürglin, 1997*), and PBC-related domains are found in some algal BELL-related TALE-HD proteins, suggesting its presence in the ancestral eukaryote (*Joo et al., 2018*). In contrast, previously characterized land plant BELL proteins possess conserved SKY and BELL domains, collectively termed POX (pre-homeobox), that mediate heterodimerization with KNOX partners (*Bellaoui et al., 2001*; *Hackbusch et al., 2005*; *Smith et al., 2002*). POX domains are either highly divergent from, or unrelated to, PBC domains. Finally, in other algal BELL-related TALE-HD proteins, neither heterodimerization domain is evident (*Joo et al., 2018*). Regardless, when tested for the dimerization potential, Archaeplastida BELL-related TALE-HD proteins interact only with MEINOX TALE-HD partners and not with other BELL-related TALE-HD proteins (*Joo et al., 2018*).

Land plants are characterized by an alternation of generations whereby complex multicellular bodies develop in both haploid (gametophyte) and diploid (sporophyte) phases of the life cycle (*Hofmeister, 1862*), and elements of the KNOX/BELL system have been implicated in regulating the land plant haploid-to-diploid transition. In the moss *Physcomitrium patens*, one subclass of KNOX genes, KNOX1, is required for proper proliferation and differentiation in the diploid body (*Sakakibara et al., 2008*; *Singer and Ashton, 2007*), while another subclass, KNOX2, acts to suppress the haploid genetic program during diploid development (*Sakakibara et al., 2013*). In addition to being expressed during sporophyte development, both classes of KNOX genes are expressed in the egg cell during gamete formation (*Sakakibara et al., 2013*; *Sakakibara et al., 2008*). The expression of KNOX genes in egg cells suggests that this gamete may correspond to the (-) gamete in *Chlamydomonas*, which also expresses a KNOX protein, implying that the male gamete in land plants might express BELL proteins, as does the (+) gamete in *Chlamydomonas*. Ectopic expression of a *Physcomitrium* BELL paralog, Pp*BELL1*, has been noted to induce the diploid genetic program in specific cell types of the gametophyte, but Pp*BELL1* expression was detected in the egg cell (*Horst et al., 2016*), rather than the male gamete, as might have been expected if it is functionally similar to *Chlamydomonas* BELL. However, expression of Pp*BELL1* has also been reported to be induced in the sperm by activation of glutamate channels, with Pp*BELL1* subsequently functionally active in the sporophyte (*Ortiz-Ramírez et al., 2017*). These observations prompted our investigation of the homologous genetic programs in the liverwort *Marchantia polymorpha*.

## Results

### *M. polymorpha* possesses phylogenetically diverse TALE homeodomain proteins

The *M. polymorpha* genome encodes nine TALE-HD-related family members: four KNOX genes and five BELL genes (*Bowman et al., 2017*; *Figure 1*), all of which are expressed in either the haploid sexual organs or the diploid sporophyte or both, with minimal or no expression detected by reverse transcription-polymerase chain reaction (RT-PCR) in the haploid vegetative thallus (*Figure 1—figure supplement 1*). Of the four KNOX genes, three are of KNOX1 subclass, but with only one encoding a HD, and one of KNOX2 subclass (*Figure 1—figure supplement 1*). The two subclasses arose via gene duplication in an ancestral charophycean alga (*Sakakibara, 2016*; *Joo et al., 2018*; *Frangedakis et al., 2017*). Expression of Mp*KNOX1* (Mp5g01600), which encodes a HD, is predominantly detected in archegoniophores and young sporophytes. In contrast, the two other KNOX1 genes (Mp*KNOX1A*, Mp4g12450; Mp*KNOX1B*, Mp2g11140) lacking a HD are predominantly expressed in antheridiophores. Phylogenetic analysis indicates that these HD-less paralogs are liverwort specific and are likely evolved in the ancestral liverwort (*Figure 1—figure supplement 1*). In contrast, expression of Mp*KNOX2* (Mp7g05320) is not detected in unfertilized reproductive organs and is expressed primarily during sporophyte development.

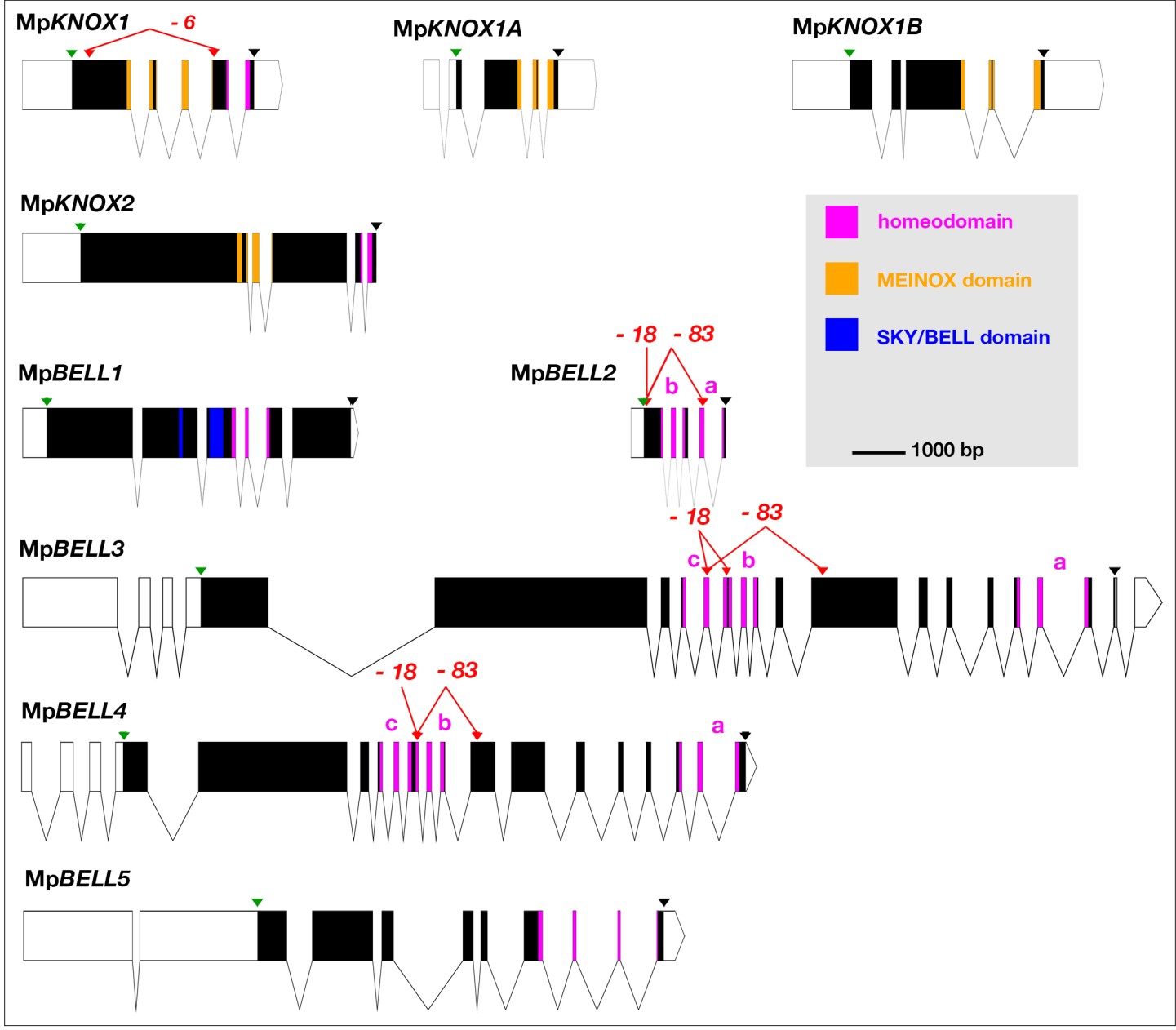

**Figure 1.** Schematic representations of the nine TALE-HD loci of *Marchantia polymorpha*. White, 5' and 3' untranslated regions (UTRs); black, coding exons; thin lines, introns; green triangles, start codon; black triangles, stop codon; red triangles, guide RNA-targeted positions. The mutant alleles generated via CRISPR-Cas9 are described in more detail in table 1; molecular lesions of some alleles can be found in *Figure 1—source data 1*. All protein annotation models are based on the *Marchantia* genome assembly of v5.1 except for Mp*KNOX2*. The Mp*KNOX2* model is based on sequences derived from reverse transcription-polymerase chain reaction (RT-PCR). In genes with multiple homeodomains, they are denoted a, b, and c. Gene models were assembled using wormweb (http://wormweb.org/exonintron).

The online version of this article includes the following figure supplement(s) for figure 1:

**Source data 1.** Molecular lesions of alleles listed in *Figure 1* and Table 1.

**Figure supplement 1.** Expression of *Marchantia polymorpha* TALE-HD genes.

**Figure supplement 2.** Unrooted Bayesian phylogram of Viridiplantae KNOX-TALE class homeodomain genes.

The *M. polymorpha* BELL genes reside in three phylogenetically distinct clades, each with its own characteristic HD (*Figure 2*; *Figure 2—figure supplement 1*). Like some algal BELL-related proteins (*Joo et al., 2018*), the presence of a PBC domain is not easily discernible in any of the predicted *M. polymorpha* BELL protein sequences. Only one *M. polymorpha* BELL gene (Mp*BELL1*, Mp8g18310)

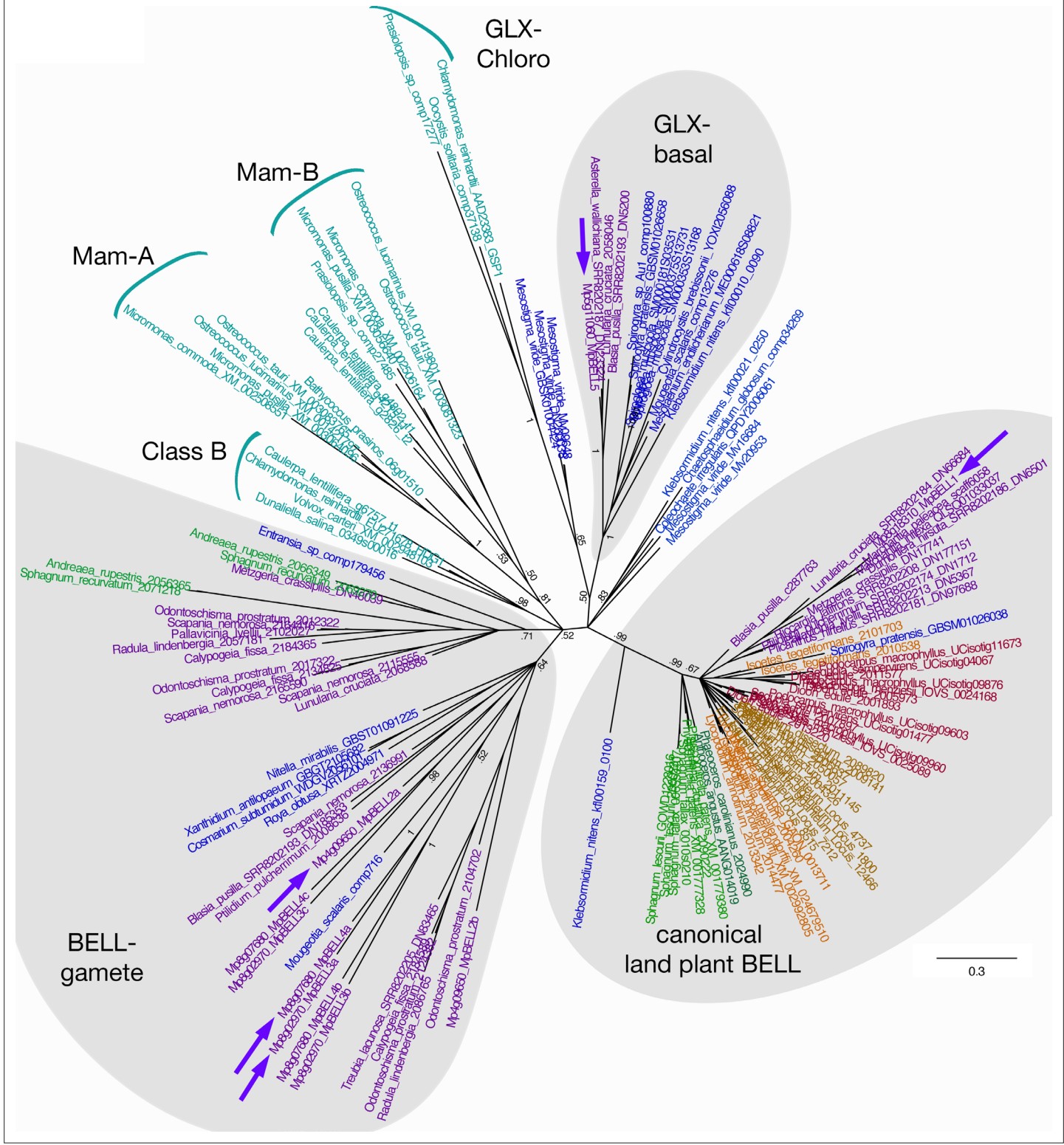

**Figure 2.** Unrooted Bayesian phylogram of Viridiplantae BELL-TALE class homeodomain genes. Tree was constructed using a nucleotide alignment of the homeodomain. The five *Marchantia polymorpha* BELL genes are highlighted (purple arrows). A clade of genes representing previously identified 'canonical' land plant BELL genes is recovered with high support. This clade includes Mp*BELL1* and charophycean algal sequences dating as far back as prior to the divergence of *Klebsormidium*. These canonical land plant BELL genes also share significant sequence similarity outside the homeodomain, including the SKY and BELL domains. The remaining BELL genes, including all chlorophyte sequences, sequences from charophycean algae, and the other four *M. polymorpha* sequences, reside in either other well-supported clades or in polytomies. Several subclades (labeled) previously identified

*Figure 2 continued on next page*

Figure 2 continued

are evident (*Joo et al., 2018*). Mp*BELL2/3/4* resides in a polytomy composed of liverwort and charophyte sequences, with other liverwort, moss, and charophyte sequences residing in a second clade; these are labeled 'BELL-gamete'. The multiple homeodomains of Mp*BELL2*, Mp*BELL3* and Mp*BELL4* are designated a, b, and c (see *Figure 1*). Mp*BELL5* resides, with other liverwort and charophyte sequences, in the GLX-basal clade. Chlorophyte, light blue; charophyte, dark blue; liverwort, purple; moss, green; hornwort, dark green; lycophyte, brown; fern, orange; seed plant, red. Numbers at branches indicate posterior probability values >50%; branches explicitly shown have probability values >50%, whereas polytomies represent nodes with probability values <50%; values within subclades are omitted for clarity.

The online version of this article includes the following figure supplement(s) for figure 2:

**Figure supplement 1.** Comparison of Viridiplantae TALE homeodomains.

is phylogenetically related to previously described, 'canonical', land plant BELL genes. Mp*BELL1* harbors a conserved canonical land plant BELL homeodomain sequence and a discernible, albeit divergent, BELL domain characteristic of other land plant BELL genes. Similar to Mp*KNOX2*, Mp*BELL1* expression is not detected in reproductive organs prior to fertilization and is expressed predominantly during sporophyte development (*Figure 1—figure supplement 1*). The other four *M. polymorpha* BELL genes fall into two distinct clades and are more closely related to algal TALE-HD genes than to the canonical land plant BELL genes (*Figure 2*). Three of the algal-like BELL genes, Mp*BELL2* (Mp4g09650), Mp*BELL3* (Mp8g02970), and Mp*BELL4* (Mp8g07680), each encode two or three homeodomains (*Figure 1*). The carboxyl HD is most conserved with those upstream becoming progressively less conserved, consistent with their origins being via successive intragenic duplications. Notably, Mp*BELL3* and Mp*BELL4* both encode large proteins, while Mp*BELL2* encodes a much smaller protein consisting of only two HDs. These three genes, named here BELL-gamete, arose via gene duplications within liverworts, and related genes are found throughout liverworts and in some moss lineages that are sister groups to the vast majority of moss diversity (*Figure 2*). The fourth algal-like sequence, Mp*BELL5* (Mp5g11060), is phylogenetically distinct from the other four *M. polymorpha* BELL genes, and resides in the previously defined 'GLX-basal' clade (*Joo et al., 2018*). Orthologs of Mp*BELL5* exist in other liverworts (*Figure 2*). The most parsimonious interpretation is that a diversity of BELL paralogs arose in a charophyte algal ancestor, with the ancestral land plant likely possessing three paralogs representing the canonical land plant, GLX-basal and BELL-gamete lineages (*Figure 2*). This diversity has persisted in *M. polymorpha* and other liverworts and, to a lesser extent, in mosses.

## Maternal Mp*KNOX1* is required for post-zygotic embryo development

During gametophyte generation, Mp*KNOX1* is expressed specifically in the egg cell but is not detected at the stage prior, when the venter canal cell is present (*Figure 3A–B*). During differentiation of the egg cell, Mp*KNOX1* appears to be expressed in a pulse early, with the signal diminishing in older archegonia (*Figure 3B*). Following fertilization, Mp*KNOX1* is expressed throughout the developing sporophyte up until the time when future sporogenous cells become distinct (*Figure 3C–E*), with expression thereafter becoming undetectable during meiotic stages (*Figure 3F*). A female harboring a loss-of-function allele consisting of a 2.3 kb deletion spanning the Mp*KNOX1* locus (*Figure 1*; Table 1) was crossed with wild-type males. In contrast to control crosses using wild-type males and females where sporophyte production was observed (*Figure 3G–I*), mature sporophytes never developed on gametangiophores from >100 crosses between wild-type males and Mp*knox1-6^{ge}* females (*Figure 3J–L*). However, on closer inspection, on rare occasions (3/129 fertilization events) did we encounter arrested multicellular embryos within senescing archegoniophores (*Figure 3—figure supplement 1*). A closer examination of female Mp*knox1-6^{ge}* mutants revealed a developmental arrest of the zygote following fertilization, with the zygote failing to undergo cytokinesis (*Figure 3K*). However, two gametophytic tissues, the calyptra and the pseudoperianth, whose growth is dependent upon fertilization (*Hofmeister, 1862*), commence development normally. In Mp*knox1-6^{ge}* mutants, as in wild type, the calyptra undergoes periclinal divisions indicating successful fertilization and implying an intergenerational (sporophyte to gametophyte) signal inducing its development (*Figure 3H–I and K–L*). Likewise, the pseudoperianth, a ring of tissue surrounding the archegonium that only develops post-fertilization, initially develops normally in Mp*knox1-6^{ge}* mutants (*Figure 3H–I and K–L*). Subsequently, both the calyptra and pseudoperianth are developmentally arrested between 1 and 2 weeks post fertilization (wpf) in Mp*knox1-6^{ge}* mutants. In addition, Mp*knox1-6^{ge}* mutants exhibit a senescent archegoniophore phenotype. Unfertilized wild-type archegoniophores remain green for multiple

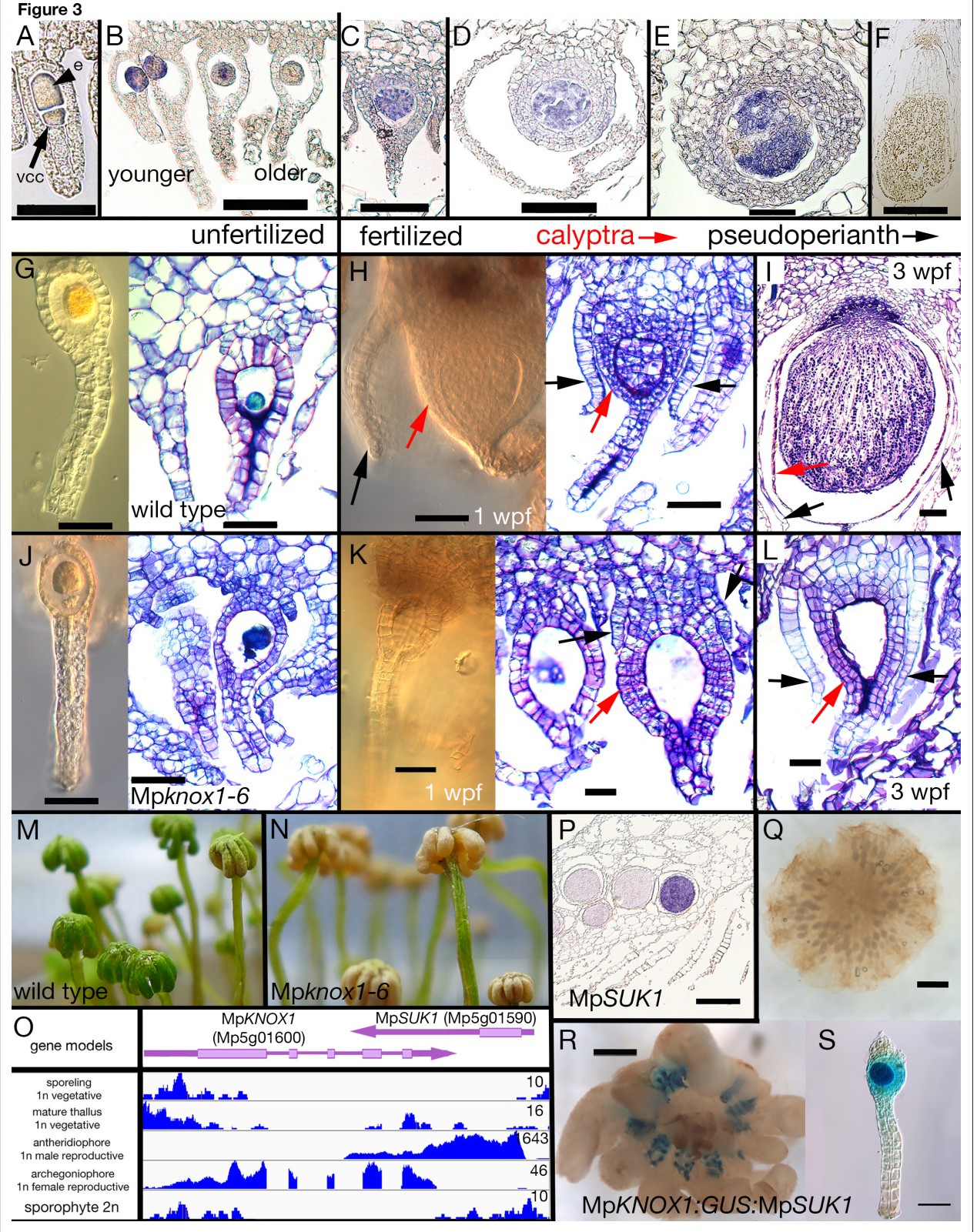

**Figure 3.** Maternal Mp*KNOX1* is required for post-zygotic embryo development. (**A–F**) Mp*KNOX1* expression pattern. Mp*KNOX1* transcripts could be detected via *in situ* hybridization in the egg (e) cells of unfertilized archegonia (**B**), but not in the venter canal cell (**A**; vcc). After fertilization, expression is detected in young developing sporophytes (approximately 1 week post fertilization (wpf); **C, D**), with expression continuing until foot and seta begin to differentiate (**E**). Mp*KNOX1* is not expressed in older sporophytes where sporogenous tissue has differentiated (**F**). (**G–L**) Comparison of wild-type

*Figure 3 continued on next page*

*Figure 3 continued*

and Mp*knox1* development; unfertilized (**G, J**), 1 wpf (**H, K**), and 3 wpf (**I, L**). Mp*knox1-6^ge* egg cells appear wild-type-like (unfertilized; **G, J**). In contrast to wild-type (**H, I**), the embryo of Mp*knox1-6^ge* mutants does not develop post fertilization (**K, L**). The zygote fails to undergo cytokinesis, with the nucleus disappearing after 3 weeks, leaving an empty space (approximately 3 wpf; **L**). The initial outgrowth of the pseudoperianth (black arrows) and calyptra (red arrows) post-fertilization is not affected in Mp*knox1-6^ge* mutants, but their development is arrested as well approximately 1 wpf (**K, L**). (**M, N**) Archegoniophores of Mp*knox1-6^ge* mutants begin to senesce approximately 2 weeks after maturation (**N**), while wild-type archegoniophores of the same age remain green (**M**). (**O**) Based on RNA-sequencing (RNA-seq) data and associated gene models, the 3' end of Mp*KNOX1* overlaps with the 3' untranslated region (UTR) of Mp*SUK1*, which is transcribed from the opposite strand. Predicted full-length Mp*KNOX1* transcripts, that is, those including exons 2 and 3, are present primarily in the archegoniophore, consistent with the *in situ* data and semiquantitative reverse transcription-polymerase chain reaction (sqRT-PCR) data in *Figure 1—figure supplement 1*. FPKM (Fragments Per Kilobase of exon model per Million mapped fragments) scales are set to a standard within each tissue to allow potential weakly expressed transcripts to be visualized. (**P**) Signal of Mp*SUK1 in situ* hybridization is detected in antheridia of the antheridiophore. (**Q, R**) Expression of *pro*Mp*KNOX1:GUS:SUK1* in an archegoniophore (**R**), an archegonium (**S**), and an antheridiophore (**Q**). Scale bar = 50 µm (**A, E, G, J, K, L, S**); 100 µm (**B, C, D, H, I**); 200 µm (**P**); 500 µm (**F, R**); 1000 µm (**Q**).

The online version of this article includes the following source data and figure supplement(s) for figure 3:

**Figure supplement 1.** Mp*knox1-6^ge* mutants rarely form embryos on senescing archegoniophores.

**Figure supplement 1—source data 1.** Developmental staging data of crosses between Mp*knox1-6* females and wild-type males.

**Figure supplement 2.** Expression profile of *pro*Mp*KNOX1* transcriptional β-glucuronidase reporter gene fusions unravels regulation by antisense transcript Mp*SUK1*.

**Figure supplement 3.** Mp*KNOX1A* and Mp*KNOX1B* lack the homeodomain and are both primarily expressed in the antheridiophores.

weeks before they senesce (*Figure 3M*). In contrast, Mp*knox1-6^ge* archegoniophores begin to senesce around 2 weeks post maturation (*Figure 3N*), suggesting that an Mp*KNOX1*-dependent signal is required for archegoniophore maintenance.

At the Mp*KNOX1* locus, a convergently transcribed gene, *SUPPRESSOR OF KNOX1* (Mp*SUK1*, Mp5g01590), is expressed predominantly in antheridia (*Figure 3O–P*), suggesting that Mp*KNOX1* may be regulated by an antisense transcript as has been described for Mp*FGMYB* (Mp1g17210) (*Hisanaga, 2019*). Consistent with this hypothesis, a transcriptional fusion of 4.6 kb 5' of Mp*KNOX1* coding sequence with a β-glucuronidase (GUS) reporter coding sequence and 3' terminator results in expression in both egg cells and antheridia (*Figure 3—figure supplement 2*). In contrast, incorporation of 2.7 kb of the Mp*SUK1* antisense transcriptional unit immediately downstream of the GUS coding sequence (*pro*Mp*KNOX1:GUS:SUK1*) results in female-specific expression (*Figure 3*; *Figure 3—figure supplement 2*). However, if a 3' terminator sequence is inserted between the GUS coding sequence and the Mp*SUK1* sequences, the antisense repression is lost, suggesting a requirement of transcriptional overlap but in a non-sequence-specific mechanism (*Figure 3—figure supplement 2*). Mp*SUK1* also encodes a 225 aa protein, including a 55 aa domain conserved across most liverworts but not found in other land plant lineages, that does not overlap with the Mp*KNOX1* coding sequence (*Figure 3O*).

In addition to the full-length Mp*KNOX1* gene, two additional KNOX1-related genes, Mp*KNOX1A* and Mp*KNOX1B*, are encoded in the *M. polymorpha* genome. Both genes encode a conserved KNOX1 MEINOX domain (*Figure 3—figure supplement 3A*) but lack a HD and are predominantly expressed in the antheridia and sperm cells (*Figure 3—figure supplement 3B-E*).

## Mp*BELL234* are expressed in antheridia, archegonia, and young embryos

The acropetal development of antheridia within antheridiophores results in the youngest being peripheral. Mp*BELL2*, Mp*BELL3*, and Mp*BELL4* expression was detected in stage 2–4 antheridia (*Higo et al., 2016*) within antheridiophores as assayed by *in situ* hybridization (*Figure 4A, C and E*). Likewise, translational Mp*BELL2* and transcriptional Mp*BELL3* and Mp*BELL4* GUS reporter gene fusion lines using 3.5 kb (Mp*BELL2*), 5.6 kb (Mp*BELL3*), and 5.8 kb (Mp*BELL4*) of sequence 5' to a transcriptional start site exhibit signals in antheridia of developing antheridiophores (*Figure 4B, D, F and G*). In these reporter lines, no signal was detected in unfertilized archegoniophores or developing sporophytes, nor was any expression observed in the vegetative gametophyte. However, in contrast to the above reporter lines, strand-specific transcriptomic analysis suggests that shorter transcripts, likely produced from an alternative promoter, are generated at the Mp*BELL3* and Mp*BELL4* loci in archegoniophores and sporophytes (*Figure 4—figure supplement 1A, B*). Indeed, expression of both

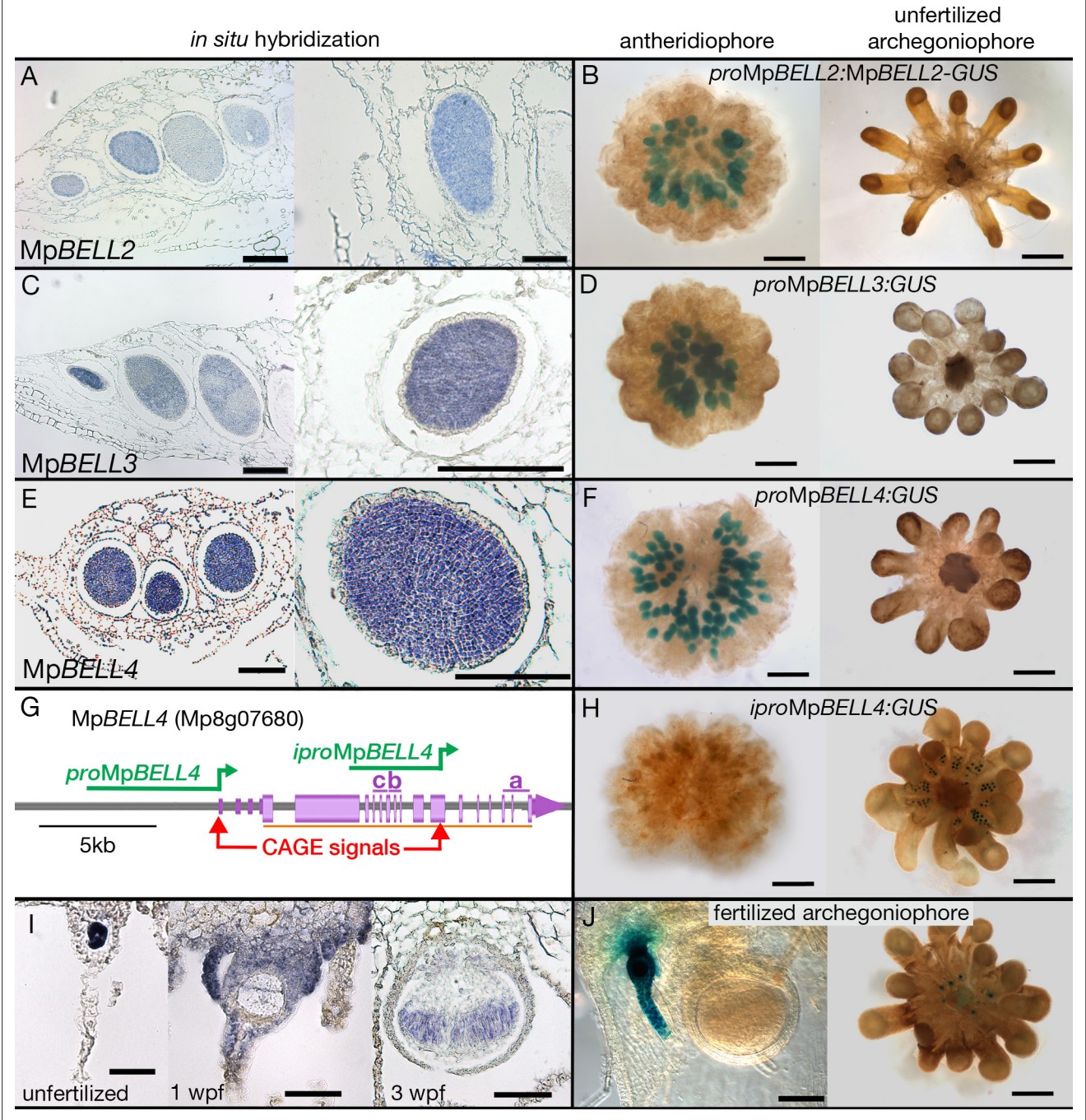

**Figure 4.** Expression patterns of Mp*BELL2*, Mp*BELL3,* and Mp*BELL4*. *In situ* localization of all three Mp*BELL* mRNAs in antheridia as seen in antheridiophore cross-sections; younger antheridia are toward the left (**A, C, E**). Antheridiophores with translational (**B**) or transcriptional (**D, F**) β-glucuronidase reporter gene fusions of Mp*BELL2* (**B**), Mp*BELL3* (**D**), and Mp*BELL4* (**F**). These reporter genes all harbor sequences 5' of the longest predicted transcript at each of the loci; for example, *pro*Mp*BELL4* in (**G**). Signal appears strongest in young- to medium-aged antheridia (stage 3 and stage 4; *Higo et al., 2016*), with the signal being lost in older antheridia toward the center of the antheridiophore (**B, F**), possibly due to draining of spermatogenous tissue. Staining was not observed in unfertilized archegonia (**B, D, F**). (**I**) *In situ* hybridization using the full-length Mp*BELL4* coding sequence exhibits signals in egg cells of unfertilized archegonia, gametophytic tissues surrounding fertilized archegonia at 1 week post fertilization (wpf), as well as sporophytes up to at least 3 wpf. (**H, J**) Reporter genes constructed using an alternative promoter internal of the Mp*BELL4* locus (*ipro*Mp*BELL4*) marked with a Cap Analysis of Gene Expression (CAGE) signal (**G**) exhibit a signal in unfertilized archegonia, but not in antheridia (**H**). Signal remains visible in developing unfertilized archegonia within the center of the archegoniophore, but no signal was observed in sporophytes at 2 wpf (**J**). Scale bar = 200 µm for left panels in **A, C, E** and the right panel in **C**, panels **B, D, F, H**, and the right panel in **J**; 100 µm for right panels in **A, E, I** and the left panel in **J**; 50 µm for the left and middle panel in **I**.

*Figure 4 continued on next page*

*Figure 4 continued*

The online version of this article includes the following figure supplement(s) for figure 4:

**Figure supplement 1.** Expression profiles of MpBELL loci.

**Figure supplement 2.** Mp*BELL3* locus is a source of multiple transcripts in antheridia versus archegonia/sporophytes.

**Figure supplement 3.** Mp*BELL4* transcripts in archegonia/sporophytes lack homeodomains.

genes was detected in egg cells of unfertilized archegonia, gametophytic tissues surrounding fertilized archegonia at 1 wpf, as well as sporophytes up to at least 3 wpf via *in situ* hybridization (*Figure 4I*; *Figure 4—figure supplement 1C*). An Mp*BELL4* reporter construct using a 4.2 kb sequence upstream of a CAGE (Cap Analysis of Gene Expression) signal (*Montgomery et al., 2020*) corresponding to the presumed alternative transcriptional start site of the archegonial/sporophytic transcript results in an egg-cell and post-fertilization gametophytic signal; however, no expression was observed in the older sporophytes (*Figure 4G–H and J*). Directed RT-PCR experiments also confirmed the existence of a shorter transcript encoding homeodomain sequences produced from Mp*BELL3* in archegoniophores and during sporophyte development (*Figure 4—figure supplement 2*).

## Paternal and maternal MpBELL are required for proper embryo development

To examine whether MpBELL234 could provide the male counterpart to the female MpKNOX1, we created loss-of-function alleles for each gene (*Table 1*). When used as male parents in crosses with wild-type females, Mp*bell2*, Mp*bell3,* and Mp*bell4* single mutants did not exhibit an aberrant phenotype; however, when Mp*bell3* Mp*bell4* (Mp*bell34*) double mutant or Mp*bell2* Mp*bell3* Mp*bell4* (Mp*bell234*) triple mutant males were used in crosses with wild-type females, mature sporophytes were formed at a reduced frequency compared to crosses between wild-type parents (*Figure 5A–B*). To quantify the reduction in fertility, we first examined crosses between wild-type parents in greater detail. A majority of fertilization events in crosses between wild-type males and wild-type females produced mature sporophytes (*Figure 5A–B*). However, nearly 40 % of wild-type crosses were either aborted at the zygote stage or arrested at a globular multicellular stage, characteristic of about 1 wpf. As we did not observe sporophytes arrested at later stages, sporophytes that surpass the 1 wpf stage must generally progress to maturity. In contrast, most fertilization events derived from crosses between wild-type females and Mp*bell(2)34* males resulted in sporophyte development that only progressed to a globular stage characteristic of the first week of wild-type development, followed by sporophyte arrest. A smaller number of fertilization events were aborted at zygote formation, resembling the phenotype observed in female Mp*knox1-6^ge* mutants (*Figure 5A–B*). However, mature sporophytes with viable spores were produced in about 17 % of fertilization events. In individual crosses in which mature sporophytes form, the corresponding archegoniophores remain viable until after the sporophytes have matured, while unfertilized archegoniophores undergo senescence. The archegoniophores with only arrested sporophytes senesced in a manner similar to unfertilized archegoniophores.

Since both Mp*BELL3* and Mp*BELL4* expression was detected in the egg cell, we performed reciprocal crosses between Mp*bell(2)34* females and wild-type males and we observed a similar distribution of sporophyte phenotypes compared to wild-type females crossed with Mp*bell234* males: aborted zygotes, aborted multicellular embryos, and phenotypically normal sporophytes (*Figure 5A–B*). However, in crosses where both the male and female harbored mutations in both Mp*BELL3* and Mp*BELL4*, no mature sporophytes were formed, with a majority of embryos arrested at the zygote stage (*Figure 5A–B*), a phenotype similar to, albeit less severe than, that observed for Mp*knox1* mutants. Similar sporophyte phenotype ratios in reciprocal crosses between Mp*knox1* and Mp*bell34* have been described independently (*Hisanaga, 2021*).

We next examined whether MpKNOX1 and MpBELL proteins could interact. We chose antheridial-expressed full-length MpBELL4 for analysis due to the truncated nature of MpBELL2 and the extreme length of MpBELL3. In a split Yellow Fluorescent Protein (YFP) BiFC assay in *Nicotiana benthamia* leaf epidermal cells, MpKNOX1 on its own was cytoplasmically localized, but when co-expressed with nuclear localized MpBELL4, MpKNOX1 signal became nuclear (*Figure 5C–F*). A similar interaction was observed with MpKNOX2 and MpBELL1 (*Figure 5—figure supplement 1*). These interactions are selective, since neither interaction between MpKNOX1 and MpBELL1 nor between MpKNOX2

**Table 1.** Alleles used in this study.
Nomenclature is as outlined previously (*Bowman et al., 2016c; Montgomery et al., 2020*).

| Name | Gene number(s) | Type | Details | Gender | Putative effect |
|---|---|---|---|---|---|
| Mpknox1-6^ge | Mp5g01600 | CRISPR 2 gRNAs | 2360 nt del | Female | Null |
| Mpbell2-1^ge | Mp4g09650 | CRISPR 1 gRNAs | 1 nt in | Female | Hypomorph |
| Mpbell2-2^ge | Mp4g09650 | CRISPR 1 gRNAs | 10 nt del | Male | Hypomorph |
| Mpbell2-11^ge | Mp4g09650 | CRISPR 1 gRNAs | 1 nt del | Male | Hypomorph |
| Mpbell3-4^ge | Mp8g02970 | CRISPR 2 gRNAs | 2217 nt del | Male | Null |
| Mpbell3-10^ge | Mp8g02970 | CRISPR 2 gRNAs | 2218 nt del | Male | Null |
| Mpbell4-5^ge | Mp8g07680 | CRISPR 2 gRNAs | 1135 nt del | Male | Null |
| Mpbell4-24^ge | Mp8g07680 | CRISPR 2 gRNAs | 1135 nt del | Male | Null |
| Mpbell34-49^ge (Mpbell3-1^ge Mpbell4-1^ge) | | CRISPR 4 gRNAs | | Male | |
| MpBELL2 | Mp4g09650 | | wt | | Wild type |
| Mpbell3-1^ge | Mp8g02970 | | 6 nt in/1 nt del; 1 nt in | | Hypomorph |
| Mpbell4-1^ge | Mp8g07680 | | 1 nt del | | Hypomorph |
| Mpbell234-15^ge (Mpbell2-15^ge Mpbell3-15^ge Mpbell4-15^ge) | | CRISPR 4 gRNAs | | Male | |
| Mpbell2-15^ge | Mp4g09650 | | 23 nt in/10 nt del | | Hypomorph |
| Mpbell3-15^ge | Mp8g02970 | | 49 nt in/1 nt del; 1 nt in | | Hypomorph |
| Mpbell4-15^ge | Mp8g07680 | | 1 nt del | | Hypomorph |
| Mpbell234-83^ge (Mpbell2-83^ge Mpbell3-83^ge Mpbell4-83^ge) | | CRISPR 6 gRNAs | | Male | |
| Mpbell2-83^ge | Mp4g09650 | | 110 nt in/1401 nt del | | Null? |
| Mpbell3-83^ge | Mp8g02970 | | 2217 nt del | | Null? |
| Mpbell4-83^ge | Mp8g07680 | | 7 nt in/1150 nt del | | Null? |
| Mpbell234-7^ge (Mpbell2-7^ge Mpbell3-7^ge Mpbell4-7^ge) | | CRISPR 4 gRNAs | | Female | |

*Table 1 continued on next page*

*Table 1 continued*

| Name | Gene number(s) | Type | Details | Gender | Putative effect |
|---|---|---|---|---|---|
| *Mpbell2-7^ge* | Mp4g09650 | | 16 nt in/8 nt del | | Hypomorph |
| *Mpbell3-7^ge* | Mp8g02970 | | 1 nt in/30 nt del; wt | | Hypomorph |
| *Mpbell4-7^ge* | Mp8g07680 | | 44 nt in | | Hypomorph |
| *Mpbell234-18^ge (Mpbell2-18^ge Mpbell3-18^ge Mpbell4-18^ge)* | | CRISPR 4 gRNAs | | Female | |
| *Mpbell2-18^ge* | Mp4g09650 | | 13 nt in/7 nt del | | Hypomorph |
| *Mpbell3-18^ge* | Mp8g02970 | | 97 nt in/304 nt del; 1 nt del | | Hypomorph |
| *Mpbell4-18^ge* | Mp8g07680 | | 1 nt del | | Hypomorph |
| *internal_proEF1:XVE>> amiR-MpE(z)1^SkmiRt66* | Mp5g18040 | Transgene | Knock-down | Both | Hypomorph |
| *proEF1:MpKNOX1* | Mp5g01600 | Transgene | Ectopic overexpression | Both | Hypermorph |
| *proEF1:XVE>> MpBELL3* | Mp8g02970 | Transgene | Inducible overexpression | Both | Hypermorph |
| *proEF1:MpKNOX1; proEF1:XVE>> MpBELL3* | Mp5g01600 | Transgene | Ectopic overexpression | Both | Hypermorph |
| *proMpBELL2:MpBELL2-GUS* | Mp4g09650 | Transgene | Translational β-glucuronidase gene fusion | Both | |
| *proMpBELL3:GUS* | Mp8g02970 | Transgene | Transcriptional β-glucuronidase gene fusion | Both | |
| *proMpBELL4:GUS* | Mp8g07680 | Transgene | Transcriptional β-glucuronidase gene fusion | Both | |
| *internal_proMpBELL4:GUS* | Mp8g07680 | Transgene | Transcriptional β-glucuronidase gene fusion | Both | |
| *proMpKNOX1:GUS* | Mp5g01600 | Transgene | Transcriptional β-glucuronidase gene fusion | Both | |
| *proMpKNOX1:GUSterm:SUK1* | Mp5g01600 Mp5g01590 | Transgene | Transcriptional β-glucuronidase gene fusion | Both | |
| *proMpKNOX1:GUS:SUK1term* | Mp5g01600 Mp5g01590 | Transgene | Transcriptional β-glucuronidase gene fusion | Both | |
| *proMpKNOX1A:MpKNOX1A-GUS* | Mp4g12450 | Transgene | Translational β-glucuronidase gene fusion | Both | |
| *proMpKNOX1B:MpKNOX1B-GUS* | Mp2g11140 | Transgene | Translational β-glucuronidase gene fusion | Both | |

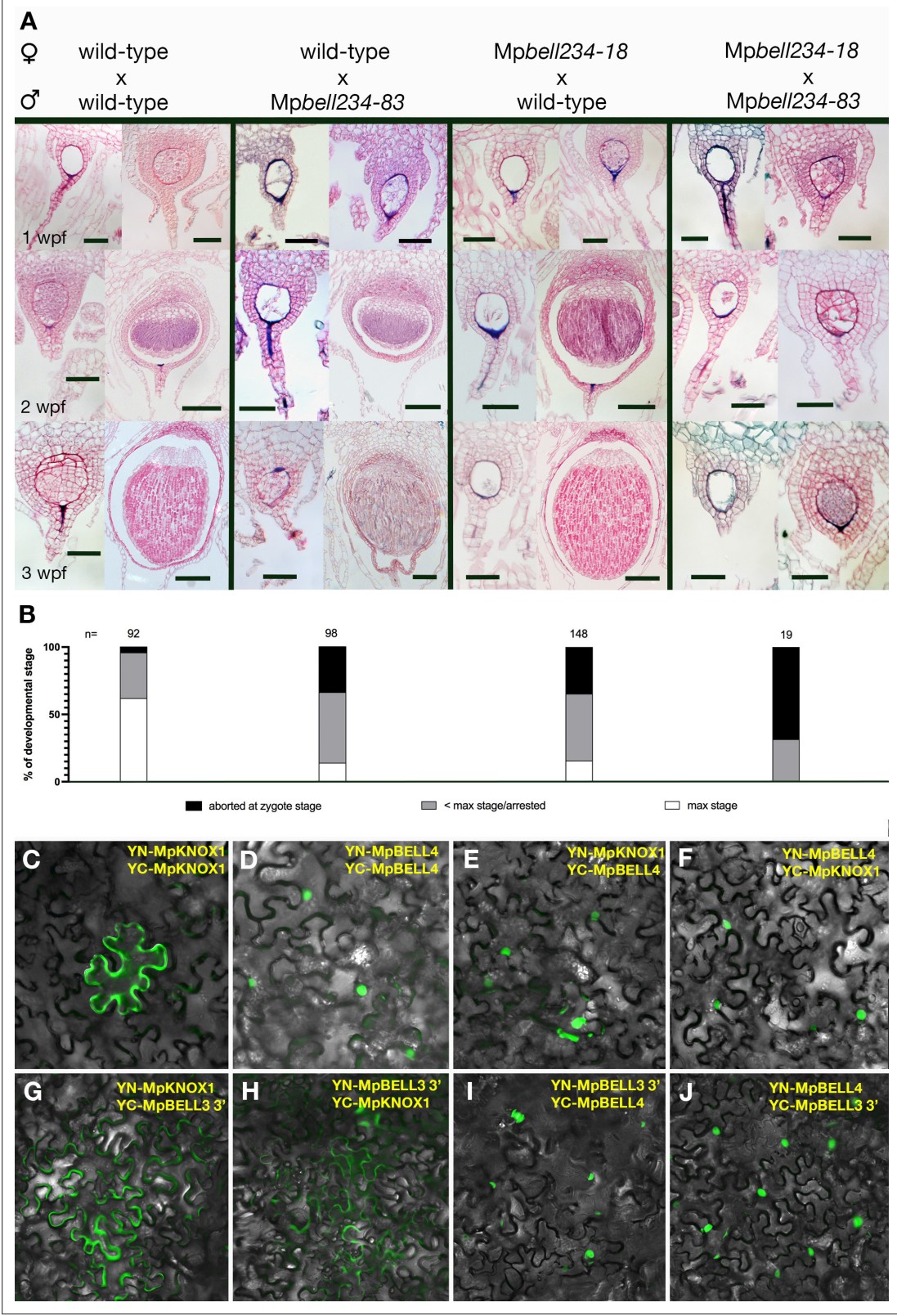

**Figure 5.** Sporophyte development entails BELL activity in both male and female gametes. (**A**) Wild-type females crossed with wild-type males (left column), reciprocal crosses between wild-type and Mp*bell* mutant alleles (middle two columns), and crosses between Mp*bell234* homozygotes (right column) observed after 1, 2, and 3 weeks post fertilization (wpf). All crosses produced aborted embryos or arrested sporophytes (left pictures in each panel). However, crosses between Mp*bell234-18*[ge] and Mp*bell234-83*[ge] only produced embryos aborted at the zygote stage or arrested at approximately

*Figure 5 continued on next page*

*Figure 5 continued*

the 1 wpf stage. See Table 1 for details on mutant alleles. (**B**) Percentages of developmental stages observed for each of the crosses. Plants were crossed once and then examined after 1, 2, or 3 wpf. Observed embryos were grouped into the following developmental stages: aborted zygotes, sporophytes that were arrested at a stage younger than the maximum stage possible, and sporophytes that had reached the maximum stage expected. N = the total number of observed fertilization events. See *Figure 5—source data 1* for details. (**C–F**) BiFC (Bimolecular fluorescence complementation) assay of protein-protein interaction of MpKNOX1 and MpBELL4 in *Nicotiana benthamiana* leaves. Homodimers of MpKNOX1 show cytoplasmic localization (**C**), while MpBELL4 homodimers localize to the nucleus (**D**). Co-expression of both proteins resulted in nuclear localization of heterodimers (**E, F**). (**G–J**) BiFC assays of the protein derived from the shorter 3′ transcript of Mp*BELL3* (*Figure 4—figure supplement 2*) with MpKNOX1 show weak cytoplasmic interaction (**G, H**). However, MpBELL4 interacts strongly with $_{short}$MpBELL3 and results in nuclear localization (**I, J**). Scale bar = 50 µm, for all left pictures in (**A**) panels, all double mutant pictures, and all 1 wpf pictures; 100 µm for right pictures of the 2 wpf panel; 200 µm for all right pictures of the 3 wpf panel.

The online version of this article includes the following figure supplement(s) for figure 5:

**Source data 1.** Developmental staging data of crosses between wild-type and Mp*bell234* mutants.

**Figure supplement 1.** MpKNOX1/MpBELL4 and MpKNOX2/MpBELL1 exhibit specificity in their interactions.

**Figure supplement 2.** Co-expression strategy to express Mp*KNOX1* and Mp*BELL3* simultaneously in the vegetative gametophyte.

and MpBELL4 was observed (*Figure 5—figure supplement 1*). In contrast to full-length MpBELL4, we did not observe any efficient interaction between the shorter sporophyte-expressed MpBELL3 and MpKNOX1 in this heterologous system (*Figure 5G–H*). Surprisingly, however, in this system, full-length MpBELL4 and sporophyte-expressed MpBELL3 interacted (*Figure 5I–J*).

Expression of the fifth *M. polymorpha* MpBELL gene, Mp*BELL5*, may be restricted to the archegoniophore among tissues examined (*Figure 4—figure supplement 1A, B*). As with Mp*KNOX1* (*Figure 3*) and Mp*FGMYB* (*Hisanaga, 2019*), the Mp*BELL5* transcript may also be opposed by a convergently transcribed locus in antheridia (*Figure 4—figure supplement 1A, B*).

## Activation of sporophyte gene expression in the vegetative gametophyte

The expression patterns and mutant phenotypes of Mp*BELL2/3/4* and Mp*KNOX1* genes are consistent with a role in activating diploid gene expression, and thus we examined whether ectopic (co-) expression of these genes in vegetative gametophyte is sufficient to activate diploid gene expression (*Figure 5—figure supplement 2A*). Ectopic expression of Mp*KNOX1* alone does not alter expression of Mp*KNOX2*, Mp*BELL1,* or Mp*BELL3* in the vegetative gametophyte (*Figure 5—figure supplement 2B*). However, ectopic expression of either Mp*BELL3* or co-expression of Mp*KNOX1* and Mp*BELL3* in the vegetative gametophyte for 72 hr is sufficient to activate both Mp*KNOX2* and Mp*BELL1,* whose expression is normally limited to sporophyte development, or, in the case of Mp*BELL1*, also after continuous far-red light induction (*Inoue et al., 2019*; *Figure 5—figure supplement 2B*). Thus, in this context, Mp*BELL3* alone can activate Mp*KNOX1* expression, a scenario perhaps reminiscent of post-zygotic activation of Mp*KNOX1* after fertilization (*Figure 3C*, *Figure 3—figure supplement 2D*).

## Discussion

### An ancestral function for TALE-HD genes in Viridiplantae

The eukaryotic life cycle alternates between haploid and diploid phases, initiated by meiosis and gamete fusion, respectively. Organisms spanning the phylogenetic diversity of eukaryotes, including ascomycete and basiomycete fungi, Amoebozoa, brown algae, and chlorophyte algae, utilize paralogous homeodomain proteins that heterodimerize following gamete fusion to initiate the diploid genetic program, lending support for the idea that this may have been an ancestral function of homeodomain proteins. Our observations in *M. polymorpha* indicate that TALE-HD proteins, specifically MpKNOX1 and MpBELL2/3/4, are initially supplied in gametes, and that this gametic expression is required for diploid sporophyte development (*Figure 6*). MpKNOX1 is absolutely required in the egg cell, as is MpBELL3/4 in either the sperm or egg. This scenario is reminiscent of that characterized in *Chlamydomonas*, wherein KNOX is supplied by the (-) gamete and BELL via the (+) gamete, and that once together in the zygote, the diploid genetic program can be activated. Thus, the basic tenets of the genetic regulation of the haploid-to-diploid transition are conserved in two widely divergent

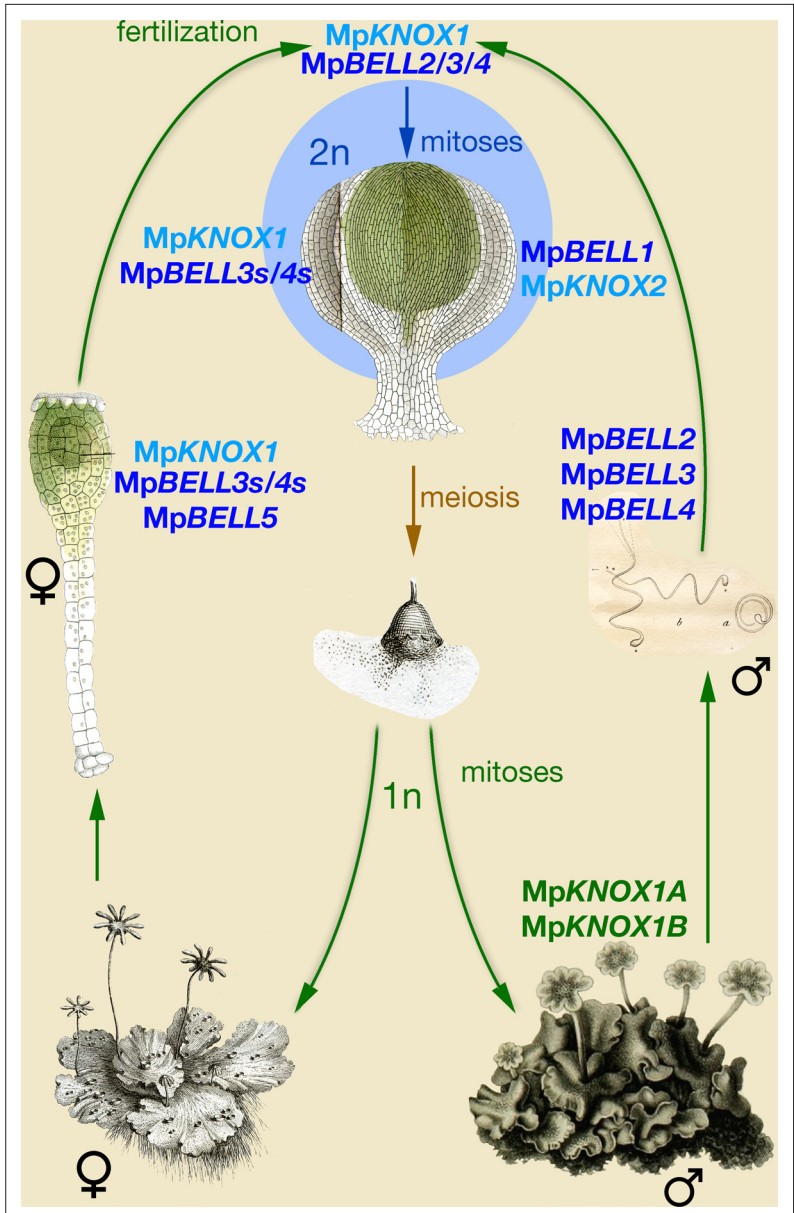

**Figure 6.** Model of KNOX and BELL function in *Marchantia polymorpha*. Three MpBELL genes (Mp*BELL2/3/4*) are expressed in antheridia in the cells that will differentiate into sperm cells. Mp*KNOX1* is expressed in the egg cell of the archegonia, along with three MpBELL genes (Mp*BELL5* and Mp*BELL3/4*). In contrast to the antheridia, where full-length transcripts are found, alternative short transcripts of Mp*BELL3/4* (Mp*BELL3s* and Mp*BELL4s*; see *Figure 4—figure supplements 2 and 3*) are produced in the egg cell. After fertilization, MpBELL3/4 derived from both sperm and egg cells and MpKNOX1 derived from the egg cell act together to activate the zygotic program. Mp*BELL1* and Mp*KNOX2* form a distinct heterodimer that acts at later stages of sporophyte development. The homeodomain lacking Mp*KNOX1A* and Mp*KNOX1B* genes expressed in the antheridia along with Mp*BELL2/3/4* potentially prevents functionality by sequestering MpBELL2/3/4 proteins. The function of Mp*BELL5* is unknown. Images are obtained from *Marchant, 1713*; *Mirbel, 1835*; *Thuret, 1851*; and *Unger, 1837*.

The online version of this article includes the following figure supplement(s) for figure 6:

**Figure supplement 1.** Phylogenetic perspective on HD gene-mediated haploid-to-diploid transitions in eukaryotes.

Viridiplantae lineages, consistent with the notion that such a system was present in the common ancestor of extant Viridiplantae.

The obvious major difference between the roles of KNOX and BELL in *Marchantia* when compared to *Chlamydomonas* is the activity of Mp*BELL34* in both the egg and sperm, rather than being confined to the sperm. Thus, questions on how the system operates mechanistically and why it might have evolved are paramount. The functions of maternal MpKNOX1 and paternal MpBELL34 could be presumed to represent the ancestral condition, with the role of maternal MpBELL34 being a derived condition. The absolute requirement for maternal MpKNOX1 (*Figure 3*) and its ability to heterodimerize with full-length MpBELL3/4 (*Figure 5*), which is expressed only paternally (*Figure 4*), suggest that this ancestral system functions in *Marchantia*, albeit with a reduced efficacy. In *Marchantia*, although neither maternal nor paternal MpBELL is absolutely required, both are necessary for efficient sporophyte development. One plausible hypothesis is that maternally supplied MpBELL3/4 evolved as a backup system to ensure diploid development following fertilization, concomitant with the evolution of sperm with reduced cytoplasmic content and highly condensed nuclei, anatomical features that could reduce the efficiency of MpBELL protein delivery by sperm. That both MpKNOX1 and $_{short}$MpBELL3 are present in the egg cell prior to fertilization and that they do not efficiently interact, at least in a split YFP BiFC assay in *Nicotiana* leaf cells (*Figure 5*), suggests that the act of fertilization may trigger a biochemical change required for their interaction. Although $_{short}$MpBELL3 can interact with full-length MpBELL4 (*Figure 5*), it is likely not critical in planta, given the absolute requirement of MpKNOX1 for sporophyte development. The requirement of both maternal and paternal MpBELL for efficient sporophyte production may also provide a mechanism to ensure activation of sporophyte development only occurs following fertilization. Finally, it is of note that a similar system, wherein two TALE-HD genes are both expressed in male and female gametes, has been described for the brown alga *Ectocarpus* (*Arun et al., 2019*). As *Ectocarpus* also possesses multicellular haploid and diploid generations and is anisogamous, perhaps similar evolutionary pressures contributed to the independent evolution of similar systems.

Given the phylogenetic affinity of mosses as a sister group to liverworts (*Renzaglia et al., 2018*), we might expect a similar KNOX/BELL program to regulate the haploid-to-diploid transition in *Physcomitrium*. The phenotype of Mp*knox1* alleles is more extreme than triple-null loss-of-function alleles of all PpKNOX1 loci where a sporophyte with viable spores forms (*Sakakibara et al., 2008*), suggesting that PpKNOX2 may also play a role in the transition in this species (*Sakakibara et al., 2013*). In *Physcomitrium*, Pp*BELL1* expression has been observed in sperm (*Ortiz-Ramírez et al., 2017*) via transcriptome analysis and in the egg via reporter gene (*Horst et al., 2016*). While the latter study reported a lack of male expression, this could be due to a loss of expression in this *Physcomitrium* accession (*Meyberg et al., 2020*), or alternatively, if the *Physcomitrium* Pp*BELL1* genomic architecture is similar to Mp*BELL3/4*, the reporter gene insertion may have excluded regulatory sequences required for alternative transcription start sites. Hence, aspects of the ancestral system regulating the haploid-to-diploid transition could be present in multiple early diverging land plant lineages.

In contrast to the situation in bryophytes, in angiosperms, paternally supplied pluripotency factors related to BABYBOOM, an AP2/ERF transcription factor, can activate the zygotic genetic program (*Khanday et al., 2019*; *Conner et al., 2017*; *Conner et al., 2015*). Furthermore, despite the extensive literature on KNOX/BELL function in angiosperms, the only potential known vestige of this ancestral system existing in angiosperms is the presence of KNOX2 in the *Arabidopsis* female gametophyte, possibly in the egg cell (*Pagnussat et al., 2007*). Thus, it seems that the ancestral function has been lost entirely in some derived land plants, as it has presumably in the metazoan lineage (*Bowman et al., 2016b*).

Since liverworts are placental organisms, with the diploid sporophyte nourished by the maternal haploid gametophyte, there is opportunity for extensive intergenerational communication. For example, that the maternal gametophytic calyptra and pseudoperianth initiate their development in Mp*knox1* mutants implicates a Mp*KNOX1*-independent non-cell autonomous zygote-derived signal activated post-fertilization. In contrast, the premature senescence of Mp*knox1* archegoniophores suggests a Mp*KNOX1*-dependent intra-gametophytic signal for the maintenance of archegonial viability. Further, the premature senescence of wild-type archegoniophores bearing only aborted embryos suggests a signal emanating from older sporophytes to maintain the viability of the maternal archegoniophore tissues until sporophyte development is completed. Finally, that aborted embryos

are found even in crosses involving only wild-type parents suggests possible maternal control over allocation of resources across multiple embryos, a phenomenon similar to maternal control over fruit set in angiosperms (*Stephenson, 1981*).

## Roles of *Mp*KNOX1 in *Marchantia* sporophyte

A key land plant innovation was the evolution of a multicellular diploid generation, the embryo, via mitoses, interpolated between gamete fusion and meiosis (*Bower, 1908*). In seed plants, ferns, and mosses, KNOX1 activity is associated with continued sporophyte cell proliferation, including sporophyte apical meristem activity (*Hay and Tsiantis, 2010*; *Sakakibara et al., 2008*; *Sano et al., 2005*). While *M. polymorpha* and other liverworts lack apical meristems during sporophyte development (*Kienitz-gerloff, 1874*), Mp*KNOX1* expression is detected throughout developing sporophytes until the inception of sporogenous cell differentiation (*Figure 3*), consistent with Mp*KNOX1* having a role in cell proliferation during *M. polymorpha* sporophyte development. Thus, while MpKNOX1 retains the ancestral function in the haploid-to-diploid transition, the lack of zygotic cell division in Mp*knox1* mutants and the Mp*KNOX1* sporophytic expression pattern suggest that neofunctionalization of KNOX1 contributed to the evolution of the embryo via stimulation of cell proliferation. That Mp*BELL3/4* are also expressed during these stages of sporophyte development (*Figure 4*; *Figure 4—figure supplements 1 and 2*, 3) suggests these genes encode the heterodimeric partners of MpKNOX1 during sporophyte development. This is supported by the subfunctionalization of MpBELL genes with respect to heterodimerization partners—MpBELL3/4 specifically heterodimerize with MpKNOX1 while MpBELL1 interacts only with MpKNOX2 (*Figure 5—figure supplement 2*). This subfunctionalization resembles that observed for KNOX/BELL interactions in angiosperms (*Furumizu et al., 2015*) but represents an independent evolutionary event since all angiosperm BELL genes are MpBELL1 orthologs.

In eukaryotic lineages in which multicellularity evolved in the diploid phase of the life cycle, genes involved in the haploid-to-diploid transition, or paralogs thereof, have been repeatedly co-opted into roles directing development in the diploid generation. Most conspicuously, despite the loss of their role in zygotic gene activation (see below), both non-TALE- (eg, Hox) and TALE-HD genes act to pattern the metazoan body (*Merabet and Mann, 2016*; *Pearson et al., 2005*; *Lewis, 1978*). Likewise, in the basidiomycete fungus *Coprinopsis*, the same HD heterodimer that initiates the diploid genetic program also directs early developmental stages in the multicellular diploid (*Kamada, 2002*). In the syncytial chlorophyte alga *Caulerpa lentillifera*, differential TALE-HD expression in the diploid body was speculated to influence differentiation of fronds and stolons, but functional data are lacking (*Arimoto, 2019*). Finally, in land plants, as the multicellular sporophyte evolved increasing complexity, KNOX/BELL genetic modules were co-opted to direct development of novel organs and tissues via regulation of cell proliferation (KNOX1) or differentiation (KNOX2) in conjunction with other gene regulatory networks (*Furumizu et al., 2015*; *Hay and Tsiantis, 2010*).

## The remaining TALE-HD genes of *M. polymorpha*

The two KNOX1-related sequences lacking homeodomains (Mp*KNOX1A* and Mp*KNOX1B*) are reminiscent of similar, albeit independently evolved, proteins in angiosperms that act as inhibitors of KNOX/BELL function by forming inactive heterodimers primarily with BELL partners (*Magnani and Hake, 2008*; *Kimura et al., 2008*). The antheridial expression of Mp*KNOX1A* and Mp*KNOX1B* prompts the hypothesis that the encoded proteins may act as a safeguard against the inappropriate activity of MpBELL in the male gametophyte. This is particularly pertinent since ectopic expression of Mp*BELL3* alone in the vegetative gametophyte can activate Mp*KNOX1*, Mp*KNOX2,* and Mp*BELL1* (*Figure 5—figure supplement 2*). The lack of an aberrant phenotype observed in lines constitutively expressing Mp*KNOX1* in male gametophytes would be consistent with this hypothesis.

In contrast to the other MpBELL genes, Mp*BELL5* transcripts are only detected in archegoniophores. Given the strong heterodimerization affinity preferences of BELL proteins for KNOX proteins (*Bellaoui et al., 2001*; *Joo et al., 2018*), the only conspicuous potential partner in *M. polymorpha* would be MpKNOX1. Hence, Mp*BELL5* might play an as-yet-undefined role in the activation of zygotic gene expression. At the Mp*BELL5* locus, a convergently transcribed locus is expressed predominantly in antheridia (*Figure 4—figure supplement 1*), suggesting Mp*BELL5* may be repressed by an antisense transcript in a manner similar to Mp*KNOX1* (*Figure 3*) and Mp*FGMYB* (*Hisanaga, 2019*). Such

male-expressed antisense transcript-mediated repression apparently provides a general mechanism for female-specific expression of autosomal genes, with sex-specific regulation ultimately being linked to the feminizing locus on the female sex chromosome (*Bowman, 2016a*; *Knapp, 1935*; *Lorbeer, 1936*).

## *Marchantia* BELL gene diversity resembles that of charophyte algae

The phylogenetic diversity of BELL genes in *M. polymorpha*, and liverworts more broadly, more closely resembles the diversity observed in charophyte algae than that of other land plants (*Joo et al., 2018*; *Lee et al., 2008*). All previously described land plant BELL genes form a single clade that evolved within the charophycean algae (*Figure 2*). The single *M. polymorpha* gene in this clade, Mp*BELL1*, is the only MpBELL gene predominantly expressed during sporophyte development. The other MpBELL genes reside in two other phylogenetically distinct clades, GLX-basal and BELL-gamete, both of which include charophyte sequences (*Figure 2*). This phylogenetic pattern suggests an ancient proliferation of BELL paralogs within the charophycean algal ancestor, with *M. polymorpha* (and other liverworts and some mosses) retaining this diversity that was subsequently lost in most other land plant lineages. The structural diversity (two or three homeodomains) of the Mp*BELL2/3/4* paralogs and the extensive sequence diversity of BELL-gamete clade genes are consistent with rapid evolution of genes involved in reproductive processes (*Swanson and Vacquier, 2002*). Finally, the expression pattern of Mp*BELL5*, a member of the GLX-basal clade that evolved early in the charophytes, also suggests an as-yet-unresolved function in reproduction. Similarly, the *Chlamydomonas* genome encodes a second BELL paralog, *HDG1*, that is expressed in both (+) and (-) gametes, suggesting a role in reproduction, but whose function is unknown.

## An ancestral function for TALE-HD genes in eukaryotes

In Viridiplantae (*Chlamydomonas* and *Marchantia*), the haploid-to-diploid transition is mediated by two TALE-HD genes. In the red alga *Pyropia yezoensis,* KNOX gene expression is detected in the diploid conchosporangium, but not in haploid thalli (*Mikami et al., 2019*). The phylogenetic distribution of KNOX and BELL subfamilies and their heterodimerization affinities (*Joo et al., 2018*) suggest that the Archaeplastida common ancestor utilized KNOX/BELL TALE-HD genes to mediate the haploid-to-diploid transition. Likewise, in the brown alga *Ectocarpus,* two TALE-HD proteins, OUROBOROS and SAMSARA, mediate the transition (*Arun et al., 2019*). In contrast, in both ascomycete and basidiomycete fungi, the haploid-to-diploid transition is mediated by heterodimerization of a TALE-HD and a non-TALE-HD protein (*Gillissen et al., 1992*; *Goutte and Johnson, 1988*; *Herskowitz, 1989*; *Hull et al., 2005*; *Kues et al., 1992*; *Spit et al., 1998*; *Urban, 1996*). In the Amoebozoa *Dictyostelium*, the two homeodomain-like proteins controlling the haploid-to-diploid transition are highly divergent, rendering the phylogenetic affinities enigmatic (*Hedgethorne et al., 2017*). As these taxa span eukaryotic phylogenetic diversity, one role of homeodomain genes in the ancestral eukaryote was to regulate the haploid-to-diploid transition—the evolution of the homeodomain in the ancestral eukaryote was associated with evolution of a novel life cycle (*Figure 6—figure supplement 1*). In arguably the most intensively studied taxon, the Metazoa, zygotic gene activation has been replaced by maternally derived pluripotency factors (*Schulz and Harrison, 2019*), but homeodomain genes have been retained.

Given the ancestral eukaryote possessed a minimum of two HD genes, including one TALE-HD and one non-TALE-HD (*Bharathan et al., 1997*; *Bürglin, 1997*; *Derelle et al., 2007*), it is an intriguing question as to whether, ancestrally, both TALE and non-TALE paralogs acted in the diploid-to-haploid transition. Either possible ancestral scenario (TALE + non -TALE or TALE + TALE) requires either a co-option of a pre-existing paralog (eg, non-TALE), or alternatively, a TALE gene duplication and transference of function to the new paralog, respectively. To resolve the ancestral condition, broader phylogenetic sampling and functional analyses across additional unicellular Bikont lineages, particularly in the Excavata and the paraphyletic sister groups of the Metazoa (choanoflagellates, Filasterea, and Ichthyosporea), might be informative (*Figure 6—figure supplement 1*). Resolution of the ancestral condition could inform whether the primary ancestral function of both TALE-HD and non-TALE-HD was regulating the haploid-to-diploid transition or whether non-TALE-HD genes had another fundamental function in ancestral eukaryotes.

## Materials and methods

### Ecotypes, plant growth, transformation, and induction

*M. polymorpha* ssp *ruderalis*, ecotype BoGa, obtained from the Botanical Garden of Osnabrueck, Germany, was used in SZ lab. Plant cultivation, transformation, and induction were performed according to *Ishizaki et al., 2008* and *Althoff, 2014*. *M. polymorpha* ssp *ruderalis*, ecotype MEL, was used in JLB lab and grown, transformed, and induced according to *Ishizaki et al., 2008* and *Flores-Sandoval et al., 2015*. In case of double transformations, sporelings were co-transformed using two constructs featuring different selectable markers. Spores were drained and plated on selection media for approximately 2 weeks and then subjected to a second round of selection before being transferred to ½ B5 plates. For induction of reproductive organs, plants were transferred to white light supplemented with far red light (735 nm; 45 µmol/m$^2$/s) on ½ B5 media supplemented with 1 % glucose.

### Genotyping, RNA extraction, cDNA synthesis, and semiquantitative reverse transcription-polymerase chain reaction

DNA was extracted using a modified protocol from *Edwards et al., 1991*. Instead of vacuum drying, the pelleted DNA was air-dried. Amplicons were directly sequenced. RNA was extracted from ≈100 mg wet weight tissue using Rneasy kit from Qiagen following the manufacturer's instructions, including an on-column Dnase I digestion. The same amount of total RNA was used for complementary DNA (cDNA) synthesis using either Superscript II (Invitrogen) (SZ lab) or Bioscript (Bioline) (JLB lab) reverse transcriptase according to the manufacturer's instructions with oligo-dT$_{15}$ primer. 5' and 3' RACE-ready cDNA synthesis was performed using SMARTScribe reverse transcriptase (Takara) according to the manufacturer's instructions but with the addition of random primer mix (NEB) to allow reverse transcription of long mRNA templates. See *Supplementary file 1*, *Table 1a* for primer sequences.

### *In situ* hybridization, sectioning, staining, and microscopy

*M. polymorpha* ssp *ruderalis*, ecotype BoGa, tissue fixation, embedding, sectioning, and hybridization with digoxigenin (DIG)-labeled antisense RNA probes were performed according to *Zachgo, 2002*. Sections were either stained in toluidine blue O (TBO) alone or counterstained with ruthenium red (RR) according to *Retamales and Scharaschkin, 2014*. Staining time for RR and TBO was 90 s and 60 s, respectively. Microscopic slides were observed using a Leica MZ16 FA microscope and pictures were taken with a Leica DFC490 camera. Plants were observed using either a Lumar dissecting microscope (Zeiss) and photographed with AxioCam HRc and AxioVision software (both Zeiss) or a Leica stereomicroscope (Leica M165 FC) and photographed with an integrated Leica DFC490 camera. See *Supplementary file 1*, *Table 1b* for probe sequences.

### Bimolecular fluorescence complementation

Coding sequences of KNOX and BELL genes were seamlessly cloned into the pGREEN derivatives (*Hellens et al., 2000*) pAMON/pSUR for N-terminal split VENUS (I152L) fusion and into pURIL/pDOX for C-terminal split VENUS (I152L) fusion (*Lampugnani et al., 2016*). All constructs feature the 35 S promoter driving the N- (pAMON/pURIL) or C-terminal (pSUR/pDOX) part of the split VENUS (I152L) fluorophore for translational fusion of the gene-of-interest in frame with the split VENUS (I152L). All constructs were transformed into *Agrobacterium tumefaciens* strain GV3101 for transfection of plant leaves (*Bracha-Drori et al., 2004*). For fluorescence microscopy, usually three, approximately 50 × 50 mm, abaxial leaf tissue fragments were examined under an Axioskop two mot plus (Zeiss) microscope and photographed using AxioCam HRc and AxioVision software. For YFP filter set 46, excitation BP 500/20; beam splitter FT 515; emission BP 535/30 was used. See *Supplementary file 1*, *Table 1c* for primer sequences.

### Overexpression and transcriptional and translational fusion constructs

Primers used for overexpression with the endogenous, constitutive Mp*EF1α* promoter (*Althoff, 2014*), inducible overexpression, and transcriptional and translational fusion constructs are shown in *Supplementary file 1*, *Table 1c*. The complete 1.8 kb Mp*KNOX1* coding sequence and a 4.8 kb fragment of the Mp*BELL3* coding sequence were amplified from cDNA and seamlessly cloned (NEBuilder, NEB) into pENTR-D (Invitrogen) via *Not*I/*Asc*I sites. The Mp*KNOX1* coding sequence was subsequently

recombined into pMpGWB403 (Addgene entry #68668) for constitutive expression, whereas the MpBELL3 coding sequence was recombined into the estrogen-inducible binary pHART XVE (*Flores-Sandoval et al., 2016*). Regulatory sequences upstream of the transcriptional start site of MpKNOX1 (4.6 kb), MpBELL3 (5.6 kb), and MpBELL4 (5.8 kb) and internal of the MpBELL4 locus (4.1 kb) were amplified and cloned into pENTR-D and subsequently recombined using LR-clonase II (Invitrogen) into pMpGWB104 (Addgene entry #68558) featuring the GUS reporter gene. To elucidate the transcriptional regulation at the MpKNOX1/MpSUK1 locus, the 4.6 kb regulatory region of MpKNOX1 was seamlessly cloned (NEBuilder, NEB) into the *Hind*III/*Xba*I site 5' of the Gateway cassette of pMpGWB401 (Addgene entry #68666). The reversely transcribed MpSUK1 locus (2.7 kb) was seamlessly cloned into either the *Sac*I site or *Asc*I site (5' or 3' of the NOS terminator). Subsequently, the GUS reporter gene was recombined using pENTR-GUS (Invitrogen). For translational fusions of MpBELL2, MpKNOX1A, and MpKNOX1B, the regulatory sequence 5' upstream of the transcriptional start site including the genomic gene locus and excluding the stop codon was amplified and cloned into pRITA, a shuttle vector featuring the GUS reporter gene and NOS terminator. MpBELL2 and MpKNOX1A were cloned into the *Kpn*I or *Sal*I/*Kpn*I site, respectively. MpKNOX1B was seamlessly cloned using Infusion cloning (Clontech). All constructs were subsequently transferred to the binary vector pHART using *Not*I.

## CRISPR/Cas9-mediated gene knock-out

Potential guide RNAs (gRNAs) were obtained and screened for off targets in the *M. polymorpha* genome using CRISPOR website (http://crispor.tefor.net/). Quadruple gRNAs were each cloned into pMpGE_En04, pBC-14, pBC-23, or pBC-34 using *Bsa*I sites. Double gRNAs were cloned into pMpGE_En04 and pBC-14 (*Hisanaga, 2019*). These guides were subsequently assembled into the pMpGE_En04 backbone using *Bgl*I sites. pMpGE_En04 were gateway cloned into pMpGE010 (*Sugano et al., 2018*; *Sugano et al., 2014*; *Hisanaga, 2019*; Addgene entry #71,536), featuring Cas9 nuclease. In case of six gRNAs, the remaining two were cloned into pMpGWB401 (*Ishizaki et al., 2015*; Addgene entry #68,666) and plants doubly transformed (*Ishizaki et al., 2008*). See *Supplementary file 1*, *Table 1d* for gRNA sequences.

## GUS staining

Generally, detection of GUS activity in a minimum of three independent lines was performed overnight at 37 °C in GUS staining solution (0.05 M NaPO4, pH 7.2, 2 mM K3[Fe(CN)6], 2 mM K4[Fe(CN)6], 0.3 % (v/v) Triton X-100) supplemented with 0.6 mg/ml b-D-glucopyranosiduronic acid (X-gluc) dissolved in N,N-dimethylformamide (DMF). Only detection of $_{pro}$MpKNOX1:GUS activity in archegoniophores was performed for 4 hr in staining solution containing 5 mM K3[Fe(CN)6] and 5 mM K4[Fe(CN)6]. Antheridiophores were cleared in 70 % (w/v) chloral hydrate in 10 % (v/v) glycerol solution.

## Phylogenetics

Predicted TALE-HD sequences were assembled from land plants, charophytes, and chlorophytes, via Genbank or additional sources as detailed below. Gymnosperm sequences were obtained from Congenie (congenie.org), fern sequences from *Equisetum* and *Azolla* transcriptomes (*de Vries et al., 2016*; *Vanneste et al., 2015*), bryophyte sequences, other than *M. polymorpha* (Phytozome) and *P. patens* (Genbank), from available transcriptomes (*Wickett et al., 2014*; *Dong et al., 2019*; *Radhakrishnan et al., 2020*), *Klebsormidium* and *Caulerpa* sequences derived from genome sequences (*Hori et al., 2014*; *Arimoto, 2019*), and other charophyte and chlorophyte sequences from Genbank and additional transcriptomes (*Cooper and Delwiche, 2016*; *Ju et al., 2015*).

Complete or partial coding nucleotide sequences were manually aligned as amino acid translations using Se-Al v2.0a11 for Macintosh (http://tree.bio.ed.ac.uk/software/seal/). We excluded ambiguously aligned sequences to produce an alignment of 231 nucleotides (77 amino acids) for 132 BELL sequences. Alignments of KNOX genes included the homeodomain, MEINOX (KNOX) and ELK domains (*Joo et al., 2018*), comprising 630 nucleotides (210 amino acids) for 124 KNOX sequences. Alignments of nucleotides and amino acids were employed in the subsequent Bayesian analysis. Bayesian phylogenetic analysis was performed using Mr Bayes 3.2.1 (*Huelsenbeck and Ronquist, 2001*; *Huelsenbeck et al., 2001*). The Bayesian analyses for the nucleotide data sets were run for 15,000,000 (BELL) or 5,000,000 (KNOX) generations, which was sufficient for convergence of the two

simultaneous runs (BELL, 0.0357; KNOX, 0.0423). In both cases, to allow for the burn-in phase, 50 % of the total number of saved trees were discarded. The graphic representation of the trees was generated using the FigTree (version 1.4.0) software (http://tree.bio.ed.ac.uk/software/figtree/). Sequence alignments and command files used to run the Bayesian phylogenetic analyses can be provided upon request.

## Acknowledgements

We thank our colleagues at the Joint Genome Institute (JGI) for their work producing the *Marchantia* genome sequence; the work conducted by the US Department of Energy (DOE) JGI, a DOE Office of Science User Facility, is supported by the Office of Science of the US DOE under contract DE-AC02-05CH11231. This work was supported by Monash University and received funding from the Australian Research Council (FF0551326, DP130100177, DP170100049, DP210101423) to JLB. SZ acknowledges support from the German Research Foundation (SFB944, P13). We thank Keisuke Inoue for providing pMpGE_En04, pBC-12, pBC-14, pBC-23, and pBC-34, and Edwin Lampugnani for BiFC vectors. We also thank Claudia Gieshoidt for *Marchantia* tissue culture support.

## Additional information

### Funding

| Funder | Grant reference number | Author |
|---|---|---|
| Australian Research Council | FF0551326 | Tom Dierschke<br>Eduardo Flores-Sandoval<br>John L Bowman |
| Australian Research Council | DP130100177 | Tom Dierschke<br>Eduardo Flores-Sandoval<br>John L Bowman |
| Australian Research Council | DP170100049 | Tom Dierschke<br>Eduardo Flores-Sandoval<br>Madlen I Rast-Somssich<br>John L Bowman |
| Deutsche Forschungsgemeinschaft | SFB944 | Tom Dierschke<br>Felix Althoff<br>Sabine Zachgo |
| Australian Research Council | DP210101423 | Tom Dierschke<br>John L Bowman |
| Deutsche Forschungsgemeinschaft | P13 | Tom Dierschke<br>Felix Althoff<br>Sabine Zachgo |

The funders had no role in study design, data collection and interpretation, or the decision to submit the work for publication.

### Author contributions

Tom Dierschke, Conceptualization, Formal analysis, Investigation, Methodology, Writing - original draft, Writing - review and editing; Eduardo Flores-Sandoval, Investigation, Methodology; Madlen I Rast-Somssich, Felix Althoff, Investigation; Sabine Zachgo, Methodology; John L Bowman, Conceptualization, Funding acquisition, Methodology, Project administration, Supervision, Writing - original draft, Writing - review and editing

### Author ORCIDs

Tom Dierschke (iD) http://orcid.org/0000-0002-3547-5765
John L Bowman (iD) http://orcid.org/0000-0001-7347-3691

### Decision letter and Author response

Decision letter https://doi.org/10.7554/eLife.57088.sa1
Author response https://doi.org/10.7554/eLife.57088.sa2

## Additional files

### Supplementary files
• Supplementary file 1. Primers and guide RNAs used in this study. *Table 1a*. Primer name, gene number, sequence, and the corresponding figure used for sqRT-PCR and RACE. *Table 1b*. Primer name, gene name, and sequence used for *in situ* probe preparation. *Table 1c*. Primer name, gene name, and sequence used for overexpression, inducible overexpression, and transcriptional and translational β-glucuronidase reporter constructs. *Table 1d*. Guide RNA sequences used to generate mutant alleles.

• Transparent reporting form

### Data availability
All data generated or analysed during this study are included in the manuscript and supporting files.

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
