## [Decision Letter]

**Acceptance summary:**

This work is an exciting and critical study that speaks to the field of evolutionary developmental biology beyond plant biology. It focuses on life cycle evolution in green plants, where the haploid-dominant life cycle in green algae evolved into the diploid-dominant one in land plants. The thorough analysis of molecular phylogeny, and a detailed investigation of developmental expression provides a comprehensive account of entire TALE class proteins, and is more than expected in a single study.

**Decision letter after peer review:**

Thank you for submitting your article "Gamete-specific expression of TALE class HD genes activates the diploid sporophyte program in Marchantia polymorpha" for consideration by *eLife*. Your article has been reviewed by 3 peer reviewers, including Sheila McCormick as Reviewing Editor and Reviewer #1, and the evaluation has been overseen by Christian Hardtke as the Senior Editor. The following individual involved in review of your submission has agreed to reveal their identity: Jae-Hyeok Lee (Reviewer #2).

The reviewers have discussed the reviews with one another and the Reviewing Editor has drafted this decision to help you prepare a revised submission.

Summary:

The manuscript of Dierschke et al., represents an exciting and critical study that speaks to the field of evolutionary developmental biology beyond plant biology. It focuses on life cycle evolution in green plants, where the haploid-dominant life cycle in green algae evolved into the diploid-dominant one in land plants. TALE class HD TF KNOX and BELL subfamilies have been implicated in the haploid-to-diploid transition in the green lineage, in the unicellular *Chlamydomonas* and in Physcomitrella. The authors therefore investigated the function of TALE class homeodomain transcription factors in the liverwort M. polymorpha. The authors use a range of molecular genetics and genomic approaches to demonstrate that MpBELL234 are expressed in male gametophyte cells (antheridia), while MpKNOX1 is expressed in the female egg cell before fertilization and during sporophyte development. They provide evidence that maternally supplied MpKNOX1 is required for later sporophyte development and that paternally supplied MpBELL234 is required for zygotic development. They found two distinct heterodimeric complexes of TALE class proteins, MpKNOX1 plus MpBELL2/3/4 and MpKNOX2 plus MpBELL1. They used BiFC to demonstrate the interaction between BELL and KNOX. Finally, they suggest that Polycomb, a repressive chromatin modifying complex, is required to repress the MpBELL1 and MpKNOX2 genetic program during gametophyte development. The thorough analysis of molecular phylogeny, and a detailed investigation of developmental expression by combining RNA-seq and ChIP-seq, provides a comprehensive account of entire TALE class proteins, and is more than expected in a single study.

Essential revisions:

We have major concerns about the ChIP-seq approach that was used to demonstrate the absence of H3K27me3 on the target loci. It appears that the ChIP-seq was not replicated, and the presence/absent of histone PTMs on the loci under study has not been verified by other means. Normally the claims made in the manuscript would need to be backed up by either another replication of the ChIP-seq experiment or with validation using ChIP-qPCR. More details are presented below. In the event that this is not possible now, the phrasing should be changed to emphasize that this needed data is missing (i.e. qualify the claims).

Furthermore, the first part of the paper (p. 5-19) is about gene structure and phylogeny and we don't get to the knockdown experiment until p. 20. As these aspects are not represented in the abstract, we suggest that this section of the paper is minimized. In the discussion the phylogeny is not discussed until the end, again giving the impression that it is not that important. The lengthy molecular phylogeny section would need to be further elaborated into a significant finding or provided as a brief supplement for improving the narrative's focus. The discussion is about 8 pages and probably could be condensed to the main points.

1. This manuscript reports that KNOX2 and BELL1 form heterodimers, whose functional study may await an analysis of their knockout plants. While the authors consider KNOX2/BELL1 unlikely to be involved in the haploid-to-diploid transition, KNOX2 in the moss, Physcomitrella, plays a critical role in preventing gametophyte development from resuming in diploid sporophytes. Thereby, a de-repressed KNOX2/BELL1 heterodimer may produce aberrant phenotypes during gametophyte development in Marchantia.

2. The authors investigated if the repression of TALE-HD (MpBELL1 and MpKNOX2) in the vegetative gametophyte occurs via PRC2. They use an inducible system with miR to knockdown MpE(z). The expectation is that E(z) knockdown will decrease PRC2 activity and hence reduce overall H3K27me3 marks within the genome. It's a rather 'brutal' approach because it is likely that many loci genome-wide will be de-repressed. But the manuscript gives no information about the overall genomic landscape after the E(z) miR knockdown. If available, describe phenotypes of the amiR-MpE(z)1 plants following 17-b-estradiol treatment in addition to the TALE gene expression. For the two experiments using transgenic plants harboring conditional amiR-MpE(z) or ectopically expressed MpKNOX1 or MpBELL3, the effects of the transgenes may differ between male and female gametophytes. Please indicate whether both or single-sex plants were analyzed.

3. Detailed analysis of MpBELL2/3/4 knockouts in the male gametophytes indicates their essential role in post-fertilization sporophyte development. However, the two observed phenotypes, zygotic, or early embryonic arrest, may be due to the expression of BELL2/3/4 from the female nucleus, as indicated by the authors (P44 L6). Since both Mpbell2/3/4 male and female plants were reported, it is reasonable to ask for the phenotypes of the sporophytes produced by Mpbell2/3/4 knockout male and female.

4. The authors state on page 21 "Compared to the overall H3K27me3 landscape in wild-type gametophytic tissue, the number and amplitude of methylation peaks are consistently reduced after 48h of down-regulation of MpE(z)1." What is meant by "consistently"? This statement needs to be backed up with data and proper statistical analysis. This requires at least one replicate experiment, and certainly no claims can be made about "amplitude" of the peaks (to compare ChIP-seq peaks quantitatively 'spike in' experiments are needed, etc., etc, – not a trivial task). One option could be to take a few dozen loci and carry out ChIP-qPCR experiments.

It is stated that MpKNOX2 and MpBELL1 show altered H3K27me3 patterns (which provides the basis for the conclusion that polycomb is required to repress BELL1 and KNOX2 but not KNOX1 and BELL234). The validation of ChIP-seq data using another replicate of ChIP-seq or ChIP-qPCR is also crucial to back up the following sentence "The three primarily gametophyte-expressed KNOX1 genes were neither marked with H3K27me3 nor induced in the amiR-MpE(z)1 background".

5. Details of the ChIP-seq data (sequencing stats, coverage, %uniquely mapped reads, n. reads, was more than one peak caller used, FRip values, n. peaks, etc). Is the GEO reference for the ChIP-seq data missing? In the methods section the authors mention MACS as a peak caller. A table should be provided with loci/peaks that have been called before/after induction.

Revisions expected in follow-up work:

Another replication of the ChIP-seq experiment or validation using ChIP-qPCR.

---

## [Author Response]

Essential revisions:We have major concerns about the ChIP-seq approach that was used to demonstrate the absence of H3K27me3 on the target loci. It appears that the ChIP-seq was not replicated, and the presence/absent of histone PTMs on the loci under study has not been verified by other means. Normally the claims made in the manuscript would need to be backed up by either another replication of the ChIP-seq experiment or with validation using ChIP-qPCR. More details are presented below. In the event that this is not possible now, the phrasing should be changed to emphasize that this needed data is missing (i.e. qualify the claims).

Similar H3K27me3 data for Marchantia polymorpha have been published in Montgomery et al., (2020 Chromatin Organization in Early Land Plants Reveals an Ancestral Association between H3K27me3, Transposons, and Constitutive Heterochromatin. Current Biology 30 573-588), including peaks at MpKNOX2 and MpBELL1 (This publication actually used our unpublished results for corroboration of their results). Thus, given we are (1) unlikely to repeat these experiments and (2) the data were only included into this story to provide an explanation of why we did not include MpKNOX2 and MpBELL1 in further analyses, we have removed all chromatin state data from the present paper. The omission is further supported by the rather minimal advancement in scientific understanding in the experiments and the monetary limitations or repeating the experiments.

Furthermore, the first part of the paper (p. 5-19) is about gene structure and phylogeny and we don't get to the knockdown experiment until p. 20. As these aspects are not represented in the abstract, we suggest that this section of the paper is minimized. In the discussion the phylogeny is not discussed until the end, again giving the impression that it is not that important. The lengthy molecular phylogeny section would need to be further elaborated into a significant finding or provided as a brief supplement for improving the narrative's focus. The discussion is about 8 pages and probably could be condensed to the main points.

We believe the phylogenetic aspect, a good fraction of which is supplemental, is important to provide the context of which genes we are examining — while MpKNOX1 is a typical land plant KNOX1 gene (updated Figure 1— figure Supplement 2), the MpBELL genes we describe are phylogenetically distinct from any land plant BELL gene that has been described previously. During the long lockdown of the pandemic, we have revisited this analysis by including more sequences from other liverworts and mosses. We can confidently state that the ancestral land plant had three BELL paralogs, only one of which was retained by flowering plants. However, all three paralogs were retained by liverworts and at least two were retained by at least some mosses (updated Figure 2). Further, sequence comparisons suggest that each of the three BELL paralogs might recognize different target sequences (new Figure 2— figure Supplement 1). Thus, we think it necessary to include such data describing which genes we are analysing prior to discussing phenotypes of loss-of-function alleles. This is now alluded to in the abstract.

1. This manuscript reports that KNOX2 and BELL1 form heterodimers, whose functional study may await an analysis of their knockout plants. While the authors consider KNOX2/BELL1 unlikely to be involved in the haploid-to-diploid transition, KNOX2 in the moss, Physcomitrella, plays a critical role in preventing gametophyte development from resuming in diploid sporophytes. Thereby, a de-repressed KNOX2/BELL1 heterodimer may produce aberrant phenotypes during gametophyte development in Marchantia.

Since MpKNOX2 and MpBELL1 are not expressed in either female or male reproductive tissues and only commence their expression after sporophyte development has progressed we did not consider these genes as candidates for the haploid to diploid transition. However, we included data on their protein-protein interactions (updated Figure5 — figure Supplement 1) to provide further evidence that they are not involved in the haploid to diploid transition as they only interact with one another, and not with MpKNOX1 or MpBELL3/4.

2. The authors investigated if the repression of TALE-HD (MpBELL1 and MpKNOX2) in the vegetative gametophyte occurs via PRC2. They use an inducible system with miR to knockdown MpE(z). The expectation is that E(z) knockdown will decrease PRC2 activity and hence reduce overall H3K27me3 marks within the genome. It's a rather 'brutal' approach because it is likely that many loci genome-wide will be de-repressed. But the manuscript gives no information about the overall genomic landscape after the E(z) miR knockdown. If available, describe phenotypes of the amiR-MpE(z)1 plants following 17-b-estradiol treatment in addition to the TALE gene expression. For the two experiments using transgenic plants harboring conditional amiR-MpE(z) or ectopically expressed MpKNOX1 or MpBELL3, the effects of the transgenes may differ between male and female gametophytes. Please indicate whether both or single-sex plants were analyzed.

These data have been removed from the new version of the manuscript, but to satisfy the reviewers' curiosity, constitutive expression of amiR-MpE(z)1 is lethal.

3. Detailed analysis of MpBELL2/3/4 knockouts in the male gametophytes indicates their essential role in post-fertilization sporophyte development. However, the two observed phenotypes, zygotic, or early embryonic arrest, may be due to the expression of BELL2/3/4 from the female nucleus, as indicated by the authors (P44 L6). Since both Mpbell2/3/4 male and female plants were reported, it is reasonable to ask for the phenotypes of the sporophytes produced by Mpbell2/3/4 knockout male and female.

In the limited time available for experimental work due to pandemic-related shutdowns, we have focussed our attention on the question posed in this comment. Furthermore, in our communications with the authors of the manuscript cosubmitted with our manuscript (Hisanaga et al.,), we have sought to reconcile differences in our results. There were two key differences between our analyses. First, in Hisanaga et al., Mpknox1 sporophytes were obtained at a moderate frequency. Second, in Hisanaga et al., Mpbell3/4 mutations affected both males and females, whereas we had only observed an effect in males. These two discrepancies were explored and are discussed below with respect to the new data that has been added to the present manuscript.

First, we explored whether some of the discrepancies may be attributed to differences in our observations due to having two very different techniques to generate sporophytes. In Hisanaga et al., the fertilization event is in vitro, allowing a detailed analysis of early events, but precluding observations beyond two weeks in these plants. In contrast, we have analysed the results of crosses in vivo. In our new analyses (new Figure 5A,B, leftmost panel) we made detailed observation of the results of crosses between wild-type males and wild-type females, which uncovered hitherto undescribed embryo abortion, mostly at about the 1-week developmental stage. We speculate that this is due to signalling within the female bearing multiple sporophytes, as such systems are common in angiosperms, where fruits are aborted if there is an 'over'-abundance of fruit set. While this is an interesting phenomenon, we did not explore it further except to acknowledge that this provides a baseline to which other crosses need to be compared.

MpKNOX1: We performed additional crosses to quantify the maternal effects Mpknox1 mutants. In no case were we able to observe any mature sporophytes, and only in rare cases was any development beyond the zygote stage observed (new Figure 3—figure Supplement 1). Thus, we believe the difference between our results and those of Hisanaga et al., are that our allele is null (large deletion), while the allele of Hisanaga et al., is a hypomorph in which translational read-through may produce functional protein at a low level. However, we made an additional observation that implies a function of MpKNOX1 maintenance of female reproductive tissues in the absence of fertilization (new Figure 3M-N).

MpBELL234: As described above for wild type and Mpknox1, we also performed a large number of reciprocal crosses between wild type and Mpbell34 mutants. We confirmed the result of Hisanaga et al., that Mpbell34 mutants have defects when in either male or female, and when Mpbell34 is mutant in both, most sporophytes abort at the zygote stage (new Figure 5AB). Thus, our results are now concordant. However, this brought up additional questions as our reporter genes for MpBELL2, MpBELL3 and MpBELL4 were expressed only in the male (Figure 4A-F). These reporters were constructed using sequences 5' to the longest transcripts produced from the loci (new Figure 4G). Thus, we performed additional experiments to detect MpBELL3 and MpBELL4 expression by *in situ* hybridization (new Figure 4H and new Figure 4— figure Supplement 2F). We then explored why our reporter genes exhibited male specific expression. Strand-specific analyses of available transcriptome data (Figure 4— figure Supplement 1B) suggested female and sporophyte expressed transcripts at both the MpBELL3 and MpBELL loci; however, these transcripts were much shorter than the transcripts expressed in males, but at least in the case of MpBELL3 would encode the carboxyl-most homeodomain (Figure 4— figure Supplement 1B). We further investigated these alternative transcripts by searching for potential alternative transcription start sites and using a combination of 5’- and 3’-RACE and rtPCR, we defined the transcripts present in the male, female, and sporophyte (new Figure 4— figure Supplement 2 and new Figure 4— figure Supplement 3). Using this data, we constructed a reporter gene using putative regulatory sequences internal to MpBELL4 (new Figure 4G). This reporter line drove expression in the female, specifically in the archegonia and egg cells (new Figure 4I), but was not expressed in sporophytes (new Figure 4J), perhaps due to some regulatory sequences lacking in our construct. These observations have been incorporated into the discussion of how TALE-HD genes in Marchantia activate the diploid genetic program.

4. The authors state on page 21 "Compared to the overall H3K27me3 landscape in wild-type gametophytic tissue, the number and amplitude of methylation peaks are consistently reduced after 48h of down-regulation of MpE(z)1." What is meant by "consistently"? This statement needs to be backed up with data and proper statistical analysis. This requires at least one replicate experiment, and certainly no claims can be made about "amplitude" of the peaks (to compare ChIP-seq peaks quantitatively 'spike in' experiments are needed, etc., etc, – not a trivial task). One option could be to take a few dozen loci and carry out ChIP-qPCR experiments.It is stated that MpKNOX2 and MpBELL1 show altered H3K27me3 patterns (which provides the basis for the conclusion that polycomb is required to repress BELL1 and KNOX2 but not KNOX1 and BELL234). The validation of ChIP-seq data using another replicate of ChIP-seq or ChIP-qPCR is also crucial to back up the following sentence "The three primarily gametophyte-expressed KNOX1 genes were neither marked with H3K27me3 nor induced in the amiR-MpE(z)1 background".

These data have been removed from the new version of the manuscript.

5. Details of the ChIP-seq data (sequencing stats, coverage, %uniquely mapped reads, n. reads, was more than one peak caller used, FRip values, n. peaks, etc). Is the GEO reference for the ChIP-seq data missing? In the methods section the authors mention MACS as a peak caller. A table should be provided with loci/peaks that have been called before/after induction.

These data have been removed from the new version of the manuscript.

Revisions expected in follow-up work:Another replication of the ChIP-seq experiment or validation using ChIP-qPCR.

This data has been removed from the new version of the manuscript.